# GLASS: GNN with Labeling Tricks for Subgraph Representation Learning

**Xiyuan Wang**[1]     **Muhan Zhang**[1,2, *]
[1]Institute for Artificial Intelligence, Peking University
[2]Beijing Institute for General Artificial Intelligence
{wangxiyuan,muhan}@pku.edu.cn

## Abstract

Despite the remarkable achievements of Graph Neural Networks (GNNs) on graph representation learning, few works have tried to use them to predict properties of subgraphs in the whole graph. The existing state-of-the-art method SubGNN introduces an overly complicated subgraph-level GNN model which synthesizes three artificial channels each of which has two carefully designed subgraph-level message passing modules, yet only slightly outperforms a plain GNN which performs node-level message passing and then pools node embeddings within the subgraph. By analyzing SubGNN and plain GNNs, we find that the key for subgraph representation learning might be to distinguish nodes inside and outside the subgraph. With this insight, we propose an expressive and scalable labeling trick, namely max-zero-one, to enhance plain GNNs for subgraph tasks. The resulting model is called GLASS (GNN with LAbeling trickS for Subgraph). We theoretically characterize GLASS's expressive power. Compared with SubGNN, GLASS is more expressive, more scalable, and easier to implement. Experiments on eight benchmark datasets show that GLASS outperforms the strongest baseline by $14.8\%$ on average. And ablation analysis shows that our max-zero-one labeling trick can boost the performance of a plain GNN by up to $105\%$ in maximum, which illustrates the effectiveness of labeling trick on subgraph tasks. Furthermore, training a GLASS model only takes $37\%$ time needed for a SubGNN on average.

## 1 Introduction

Graph is a natural tool for modeling objects with complex internal relationships, which is widely used in fields such as natural language processing (Yao et al., 2019), biology (Fout et al., 2017), and social network (Chen et al., 2018). Among the various graph representation learning methods, GNN has achieved state-of-the-art performance on almost all sorts of tasks. Existing GNNs are mainly designed for node (Dabhi & Parmar, 2020; Chen et al., 2020), edge (Singh et al., 2021; Galkin et al., 2021) and whole graph (Ying et al., 2021b; Yang et al., 2020) property prediction tasks. An ordinary GNN produces embeddings of a node by aggregating the features from the (multi-hop) neighbors of the node, which is equivalent to encoding a breadth-first-search (BFS) tree rooted in the node (Xu et al., 2019). Such embeddings can be used to predict node properties directly. As for edge and graph tasks, pooling the embeddings of nodes related to the structure is a prevailing method.

Though node, edge, and graph tasks are the three most common graph representation learning tasks, properties of subgraphs are also worth predicting. Take company structure network as an example. The nodes are employees, and the edges between them represent cooperation relations. We want to predict the performance of a department, in other words, a subgraph in the network. Obviously, on the one hand, we need to consider the internal organization (such as the collaboration within the department and the competence of individual employees). If the structure is disorganized or the employees are incompetent, we can expect the department to perform poorly. On the other hand, the external information of the department also deserves attention. A department is less likely to be productive if the company as a whole is facing bankruptcy. In contrast, close cooperation with

---

*Corresponding author: Muhan Zhang

other remarkable departments can be a sign of good performance. As illustrated by this example, a subgraph task is to predict the property of subgraphs in the whole graph. It needs to consider the topology both inside and outside the subgraph. It may also need to combine multi-level information of nodes, edges, higher-order substructures, and even the whole graph. Thus, a very general model is needed, and a natural idea is to extend ordinary GNNs to subgraph tasks. Figure 1 left shows a typical subgraph to predict. The target subgraph $\mathcal{S}$ is embedded in the whole graph and may have multiple connected components, and our task is to produce a subgraph representation, which can be used to predict properties of $\mathcal{S}$.

However, in experiments, we find that SubGNN (Alsentzer et al., 2020), the current state-of-the-art method for subgraph tasks, only slightly outperforms plain GNNs which directly pool node embeddings within the subgraph as the subgraph representation. SubGNN replaces the message passing between nodes with a subgraph-level message passing framework and designs three channels. Each channel is further divided into an internal and a border module to aggregate subgraph features separately. Despite its performance, SubGNN is both space and time-consuming due to its lengthy precomputation. Furthermore, the units of message passing are subgraph patches sampled from the whole graph randomly, and there is no guarantee of the optimality of the samples, leading to high variance in performance and dubious robustness. Last but not least, the framework of SubGNN is overly complicated—many of its designs seem ad hoc or suboptimal, resulting in poor compatibility with recent advances in GNN research. Nevertheless, by comparing SubGNN with plain GNNs, we find that differentiating internal and external topology is crucial for subgraph tasks. Inspired by this insight, we introduce a max-zero-one labeling trick (Zhang et al., 2021), which explicitly marks whether a node is within a subgraph or not, to augment GNNs and show that plain GNNs with this labeling trick are superior to SubGNN.

Here we give a brief introduction to labeling trick. First proposed by Zhang et al. (2021), labeling trick is a theoretical framework for using graph neural networks to produce multi-node representations, which show that producing expressive representations for high-order structures needs to capture the interaction among the different nodes within the structure. This theory shows that directly aggregating node representations to represent high-order structures is not expressive enough, and labeling trick can aid this problem. In implementation, a labeling trick assigns a label to each node and combines the node features and the labels as the new input node features to GNNs. Labeling trick has achieved great successes on graph representation learning in previous works. For example, the state-of-the-art link prediction method SEAL (Zhang & Chen, 2018) gains better performance with a carefully designed labeling trick. IDGNN (You et al., 2021) differentiates one center node from others, and Distance Encoding (Li et al., 2020) uses the distance to the target nodes to label other nodes, both of which gain improved performance on node, edge, and graph tasks. In this work, we for the first time introduce labeling trick to subgraph problems, and design an expressive and scalable labeling trick called max-zero-one. Max-zero-one is the first labeling trick that enables jointly predicting a batch of structures within the same graph.

**Present work** We propose GLASS (GNN with LAbeling trickS for Subgraph), a novel and simple graph neural network for subgraph tasks. To the best of our knowledge, GLASS is the first subgraph representation learning method using the ordinary message passing framework and a labeling trick. GLASS is more scalable, more expressive, and easier to implement and extend than the existing state-of-the-art method. Theoretically, we prove that GLASS is more expressive than plain GNNs, and can capture a range of important subgraph properties defined in SubGNN like density, cut ratio, border, positions, etc. Experiments on eight datasets show that GLASS achieves new state-of-the-art performance. On synthetic datasets, GLASS with a single message passing layer beats SubGNN with three carefully designed channels by up to $48.6\%$, illustrating the expressive power of node-level message passing augmented by labeling trick for subgraph tasks. On real-world datasets, GLASS also outperforms the strongest baseline SubGNN by up to $14.3\%$. Moreover, training a GLASS model on average only takes $37\%$ time needed to train a SubGNN. With the strong performance of GLASS, our work proves the effectiveness of labeling trick on subgraph tasks.

## 2 RELATED WORK

**Subgraph Representation Learning.** Though some works have utilized subgraphs to perform other graph representation tasks (Sun et al., 2021; Wang et al., 2021; Huang & Zitnik, 2020) or

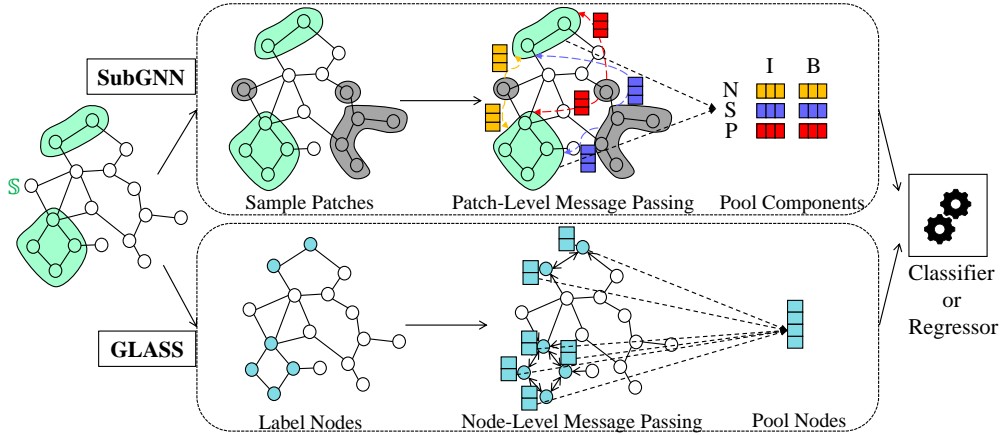

Figure 1: Subgraph tasks are to predict the properties of subgraphs in the whole graph. $\mathcal{S}$ (green) is the subgraph we want to predict. SubGNN samples anchor patches (grey), aggregates features of anchor patches to connected components of the subgraph through six independent channels (red, blue, and yellow), and pools component embeddings as the subgraph representation. GLASS labels nodes, passes messages between them, and pools node embeddings as the subgraph representation.

studied some specific tasks involving subgraphs (Bordes et al., 2014; Meng et al., 2018; Ying et al., 2020), few works have studied the general subgraph representation learning problem. Alsentzer et al. (2020) introduced the problem formally and proposed SubGNN (Alsentzer et al., 2020), the current state-of-the-art method, which samples patches from the whole graph and aggregates their features to produce subgraph representations. Before that, Sub2Vec (Adhikari et al., 2018), designed for graph classification and community detection, samples random walks in subgraphs and feeds them to the language model Paragraph2vec (Le & Mikolov, 2014) to generate embeddings for subgraphs.

**Labeling trick.** SEAL (Zhang & Chen, 2018) first introduces labeling trick to graph representation learning and applies them to link prediction. IDGNN (You et al., 2021) uses different message passing parameters for a target node and the other nodes, which is essentially a labeling trick assigning different labels to the target node and others. Distance Encoding (Li et al., 2020) uses distances to target nodes as node labels. Zhang et al. (2021) give a theoretical analysis of labeling trick and prove that they can produce the most expressive representations for substructures with a GNN expressive enough. These previous methods ignore subgraph tasks and have poor scalability due to the relabeling for every target substructure to predict. Besides these deterministic labeling methods, rGIN (Sato et al., 2021) assigns a random vector to each nodes as its label in each forward process. Similarly, GNN-RNI (Abboud et al., 2021) randomly initializes node embeddings and can approximate any functions mapping graphs to real numbers. Despite the theoretical power, random labels suffer from slow convergence and subpar performance. We also discuss other structural encoding methods in Appendix A.1. Our GLASS is an application of labeling trick to subgraph tasks. Its success verifies the theory of using GNNs and labeling trick to produce multi-node representations.

## 3 PRELIMINARIES

Let $\mathcal{G} = (\mathbb{V}, \mathbb{E}, \boldsymbol{X})$ denote a graph with a finite node set $\mathbb{V} = \{1, 2, ..., n\}$, an edge set $\mathbb{E} \subseteq \mathbb{V} \times \mathbb{V}$ and node feature matrix $\boldsymbol{X}$, whose $i^{\text{th}}$ row $X_i$ is the feature of node $i$. $N(v)$ refers to the set of nodes adjacent to node $v$. $\mathcal{S} = (\mathbb{V}_\mathcal{S}, \mathbb{E}_\mathcal{S}, X_\mathcal{S})$ is a subgraph of $\mathcal{G}$ if $\mathbb{V}_\mathcal{S} \subseteq \mathbb{V}$ and $\mathbb{E}_\mathcal{S} \subseteq (\mathbb{V}_\mathcal{S} \times \mathbb{V}_\mathcal{S}) \cap \mathbb{E}$ and $\boldsymbol{X}_\mathcal{S}$ is the stack of the rows of $\boldsymbol{X}$ corresponding to nodes in $\mathbb{V}_\mathcal{S}$. In this paper, we focus on induced subgraphs, whose edge set $\mathbb{E}_\mathcal{S} = (\mathbb{V}_\mathcal{S} \times \mathbb{V}_\mathcal{S}) \cap \mathbb{E}$. Let $\mathcal{S} \subseteq \mathcal{G}$ donate that $\mathcal{S}$ is a subgraph in $\mathcal{G}$.

**Problem (Subgraph Representation and Property Prediction).** Given the whole graph $\mathcal{G}$ , its subgraphs $\mathbb{S} = \{\mathcal{S}_1, \mathcal{S}_2, ..., \mathcal{S}_n\}$ and their target properties $\boldsymbol{T} = \{\boldsymbol{t}_{\mathcal{S}_1}, \boldsymbol{t}_{\mathcal{S}_2}, ..., \boldsymbol{t}_{\mathcal{S}_n}\}$, the goal is to learn a representation vector $\boldsymbol{h}_{\mathcal{S}_i}$ that can be used to predict $\boldsymbol{t}_{\mathcal{S}_i}$ of $\mathcal{S}_i$.

**A Plain GNN for Subgraph Tasks.** Message passing neural network (MPNN) (Gilmer et al., 2017) is a common framework of GNNs. A message passing layer aggregates embeddings from neighbors

Table 1: Six properties in SubGNN. $\mathcal{S}$ in graph $\mathcal{G}$ is a target subgraph.

| channel | I | B |
|---|---|---|
| P | Distance between connected components of $\mathcal{S}$ | Distance between $\mathcal{S}$ and the rest of $\mathcal{G}$ |
| N | Identity of internal nodes | Identity of border nodes |
| S | Internal connectivity | Border connectivity |

Table 2: Implementation of each channel in SubGNN for a target subgraph component $\mathcal{S}^{(c)}$. $N_k(\mathcal{S}^{(c)})$ means the union of the k-hop neighborhood of nodes in $\mathcal{S}^{(c)}$. $rw_I$ refers to random walk in an anchor patch $\mathcal{A}$ and $rw_B$ means random walk in the border of $\mathcal{A}$. SubGNN feeds the sampled node feature sequences into a bidirectional LSTM. $dtw(\mathcal{S}^{(c)}, \mathcal{A})$ is the normalized dynamic time warping measure (Mueen & Keogh, 2016) between the sorted node degree sequence of $\mathcal{S}^{(c)}$ and $A$. $d(\mathcal{S}^{(c)}, \mathcal{A})$ is the average shortest path between $\mathcal{A}$ and nodes in $\mathcal{S}^{(c)}$.

| channel | property | patch sampler | patch representation | similarity |
|---|---|---|---|---|
| P | I | node in $\mathcal{S}$ | node embedding | $1/(d(S^{(C)}, A) + 1)$ |
|   | B | node out of $\mathcal{S}$ | | |
| N | I | node in $\mathcal{S}^{(c)}$ | node embedding | $1$ |
|   | B | node in $N_k(\mathcal{S}^{(c)})$ | | $1/(d(S^{(C)}, A) + 1)$ |
| S | I | connected components | $rw_I$ + LSTM | $1/(dtw(S^{(C)}, A) + 1)$ |
|   | B | | $rw_B$ + LSTM | |

to update the representation of a node. The $k^{\text{th}}$ message passing layer can be formulated as follows.

$$\boldsymbol{a}_v^{(k)} = \text{AGGREGATE}^{(k)}(\{\boldsymbol{h}_u^{(k-1)} | u \in N(v)\}), \tag{1}$$

$$\boldsymbol{h}_v^{(k)} = \text{COMBINE}^{(k)}(\boldsymbol{h}_v^{(k-1)}, \boldsymbol{a}_v^{(k)}), \tag{2}$$

where $\boldsymbol{h}_v^{(k)}$ is the embedding of node $v$ at the $k^{\text{th}}$ layer and $\boldsymbol{h}_v^{(0)} = X_v$.

The embeddings at the last layer can be used to predict node properties. As for edge or graph tasks, pooling the multiset of embeddings of nodes belonging to the edge or within the graph is a widely used method. Naturally, we can extend it to subgraphs, and the representation of a subgraph $\mathcal{S}$ can be learned by pooling the embeddings of nodes within the subgraph as follows, which we call a *plain GNN*.

$$\boldsymbol{h}_\mathcal{S} = \text{READOUT}(\{\boldsymbol{h}_u | u \in \mathbb{V}_\mathcal{S}\}). \tag{3}$$

## 4 A COMPARISON BETWEEN SUBGNN AND PLAIN GNNS

In this section, we first illustrate the failure of plain GNNs on some simple subgraph tasks. Then, to find the reason for the deficiency, we dive into the state-of-the-art subgraph representation learning method, SubGNN, and compare it with plain GNNs. We find that the key advantage of SubGNN comes from dealing with internal and border topology separately, which motivates our GLASS.

Though numerous works have proven plain GNNs' graph representation learning ability, we find that they can fail in some simple subgraph representation situations. For example, the representations of two different subgraphs $\mathcal{S}$ and $\mathcal{S}'$ in Figure 2 must be the same as all nodes in the graph have identical neighborhood structures and thus equal embeddings from plain GNNs. The failure to differentiate $\mathcal{S}$ and $\mathcal{S}'$ suggests the limitation of using plain GNNs for subgraph representation learning.

Unlike plain GNNs, the existing state-of-the-art method, SubGNN, designs a special message passing architecture. SubGNN introduces three channels, namely *position*, *neighborhood*, and *structure* (P, N, S for short), and learns each channel's *internal* and *border* (I, B for short) properties separately. Table 1 defines the six properties. They claim that these six properties are key for learning powerful subgraph representations. To capture channel $i$ for a target subgraph $\mathcal{S}$, SubGNN samples anchor patches $\mathbb{A}_i = \{\mathcal{A}_i^{(1)}, ..., \mathcal{A}_i^{(n_A)}\}$ and then learns representations of each connected component in $\mathcal{S}$ by propagating messages from the anchor patches to components of $\mathcal{S}$. See the top half of

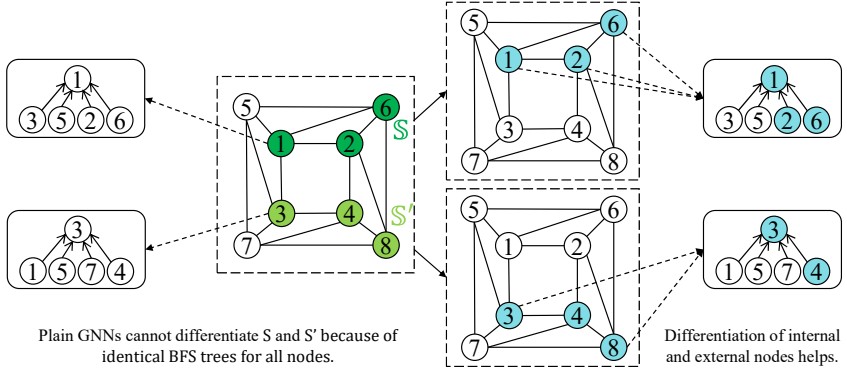

Figure 2: Plain GNNs are not enough to capture subgraph topology. For example, given a graph $\mathcal{G}$ with empty node feature, GNN representations of subgraph $\mathcal{S}$ and $\mathcal{S}'$ must be the same because of the homogeneous rooted subtree structures, though these two subgraphs are non-isomorphic. However, differentiation of nodes in and outside helps GNNs generate different embeddings for $\mathcal{S}$ and $\mathcal{S}'$. Initially, GNNs can only learn that each node in $\mathcal{S}$ and $\mathcal{S}'$ has four neighbors. However, with the differentiation, GNNs can learn that all nodes in $\mathcal{S}$ have two neighbors inside and two outside, while in $S'$, two nodes, $3$ and $8$, have one neighbor inside and three outside.

Figure 1. The $k^{\text{th}}$ subgraph-level message passing layer for channel $i$ can be formulated as follows.

$$\boldsymbol{a}_{i,\mathcal{S}^{(c)}} = \sum\nolimits_{\mathcal{A}_i \in \mathbb{A}_i} \gamma_i(\mathcal{S}^{(c)}, \mathcal{A}_i) \boldsymbol{g}_{\mathcal{A}_i}, \tag{4}$$

$$\boldsymbol{h}_{i,\mathcal{S}^{(c)}}^{(k)} = \sigma(W_i \cdot [\boldsymbol{a}_{i,\mathcal{S}^{(c)}}, \boldsymbol{h}_{i,\mathcal{S}^{(c)}}^{(k-1)}]), \tag{5}$$

where $\mathcal{S}^{(c)}$ is the $c^{\text{th}}$ component of the target subgraph $\mathcal{S}$, $\gamma_i(\mathcal{S}^{(c)}, \mathcal{A}_i)$ is a similarity function between $\mathcal{S}^{(c)}$ and an anchor patch $\mathcal{A}_i$, $\boldsymbol{g}_{\mathcal{A}_i}$ is a (pretrained) representation of patch $\mathcal{A}_i$, $\boldsymbol{h}_{i,\mathcal{S}^{(c)}}^{(k)}$ is the representation of the component $\mathcal{S}^{(c)}$ in channel $i$ at the $k^{\text{th}}$ layer, and $\sigma$ is an activation function and $W_i$ is a weight matrix. The implementation of each channel in SubGNN can be found in Table 2. The representation of the whole subgraph is the sum of the representations of all components. We call the above formulation subgraph-level message passing to distinguish it from the node-level message passing defined in Equations (1) and (2).

In general, SubGNN pretrains a plain GNN to produce node embeddings, pools them to produce patch embeddings and then smoothes the embeddings of the target component with the embeddings of patches close to or structurally similar to the target component. Through the carefully designed subgraph-level message passing layer for each of the six properties separately, SubGNN can capture the internal structure, border connectivity, and position relative to the rest of the graph. Looking back on plain GNNs, we wonder why they are inferior to SubGNN. By encoding the topology of BFS trees, plain GNNs can easily capture (multi-hop) neighborhood and the position of nodes in the whole graph. However, from the example in Figure 2 we can see that plain GNNs cannot represent internal structure and border connectivity well, as they cannot tell whether a neighbor of a node is **in the subgraph or in the rest of the graph**. Therefore, the missing part of plain GNNs from SubGNN is that SubGNN passes internal and border messages separately, while plain GNNs cannot distinguish nodes in and outside the target subgraph when passing messages between nodes.

## 5  GLASS: GNNS WITH LABELING TRICKS FOR SUBGRAPH

Inspired by SubGNN, we aim to differentiate nodes in the target subgraphs from the rest of the graph and see how such differentiation can improve plain GNNs. Still using $\mathcal{S}$ and $\mathcal{S}'$ in Figure 2 as an example, the initially identical rooted trees become different when we differentiate internal and external nodes for each subgraph (Figure 2 right), leading to different node embeddings between the two subgraphs. Therefore, GNNs can differentiate these two subgraphs after the pooling. Inspired by this insight, we propose GLASS, which uses labeling trick to enhance plain GNNs.

## 5.1 THE ZERO-ONE LABELING TRICK FOR SUBGRAPHS

Introduced by Zhang et al. (2021), *zero-one labeling trick* is a general labeling trick for all node-sets. We restate it for subgraphs.

**Definition 1.** *Given a graph $\mathcal{G}$ and a subgraph $\mathcal{S}$ in $\mathcal{G}$, the **zero-one label** of node $v$ is*

$$\boldsymbol{l}_v^{(\mathcal{S})} = \begin{cases} 1 & \text{if } v \in \mathbb{V}_{\mathcal{S}} \\ 0 & \text{if } v \notin \mathbb{V}_{\mathcal{S}} \end{cases} \tag{6}$$

GLASS is a plain GNN augmented by the zero-one labeling trick. It assigns nodes in and outside the subgraph different labels and enhances the hidden representation of a node in each message-passing layer with its zero-one label. Finally, the representation of $\mathcal{S}$ is produced by pooling the embeddings of nodes in $\mathcal{S}$. The bottom half of Figure 1 illustrates the GLASS framework.

The following proposition shows that GLASS is more expressive than plain GNNs.

**Proposition 1.** *Given a graph $\mathcal{G}$ and subgraphs $\mathcal{S}$ and $\mathcal{S}'$ in $\mathcal{G}$, if plain GNNs can distinguish them, GLASS can also produce different representations of $\mathcal{S}$ and $\mathcal{S}'$. However, there exist pairs of subgraphs that plain GNNs cannot differentiate while GLASS can.*

In addition, we prove that GLASS can precisely predict two critical metrics used in SubGNN to evaluate a model's expressive power for subgraph representation learning, namely cut ratio and density, while plain GNNs cannot.

**Theorem 1.** *Given any graph $\mathcal{G}$, there exists a GLASS model that can precisely predict the density and cut ratio of any subgraph in $\mathcal{G}$.*

Now we have seen that GLASS is a more powerful model than plain GNNs for subgraph representation learning. We next compare GLASS with the state-of-the-art model, SubGNN, and analyze their expressive power differences. We show that GLASS can represent all the channels of SubGNN.

**Proposition 2.** *GLASS can cover the six properties of SubGNN defined in Table 1.*

The concrete analysis and proofs of Proposition 2 can be found in Appendix A.4. In summary, GLASS can learn all properties designed by SubGNN while using node-level message passing only.

In Appendix A.5, we further characterize the expressive power of GLASS using the theoretical framework introduced by Zhang et al. (2021). We **prove** that (1) With a GNN expressive enough, GLASS can learn the most expressive structural representations for subgraphs, thus enabling learning any functions over subgraphs (in theory can solve any tasks over subgraphs). (2) However, the expressive power of practical GNNs is usually bounded by the 1-dimensional Weisfeiler-Lehman (1-WL) test (Weisfeiler & Leman, 1968). We prove that the zero-one labeling trick can also boost those GNNs as powerful as 1-WL for subgraph tasks.

## 5.2 THE MAX-ZERO-ONE LABELING TRICK

Though the zero-one labeling trick can boost GNNs, it makes batch training hard, as the input node feature vectors change with different target subgraphs. Time-consuming message passing needs to be done separately for each target subgraph, resulting in higher time complexity than plain GNNs.

To alleviate this issue, we can combine the labels of different target subgraphs and produce their representations in a single forward process. More specifically, we can jointly label a batch of subgraphs within the graph, perform message passing on the whole graph, and then pool individual node embeddings to produce multiple subgraph representations at the same time. This enables learning a batch of subgraph representations from the same labeled graph instead of learning one subgraph representation from one labeled graph.

However, node labels might conflict with each other for different subgraphs, resulting in inconsistencies with the canonical way. Nevertheless, we argue that such processing has a controllable effect on the final results. On the one hand, as ordinary $k$-layer GNNs only encode neighbors within $k$ hops, the representation will not change if the node labels of neighbors within $k$ hops stay the same. We can expect the errors to be negligible if the change in node labels is distant from the target subgraph.

This can be achieved if the subgraphs are sparsely located in the graph. On the other hand, such inconsistencies provide a regularization effect, which prevents GNNs from overfitting the node labels of individual subgraphs but promotes learning general node features useful for multiple subgraphs.

How can we combine the node labels for different target subgraphs? One solution is to concatenate them, which seems to have no loss of information. However, this method leads to variable lengths of node labels, and depending on which other subgraphs attend this batch, it leads to possibly different node labels for the same target subgraph. The sum of all labels has similar problems. However, using the max of all labels avoids such a problem, which can help the model converge faster and facilitate processing samples in varied batch sizes. Taking the zero-one labeling trick as an example, we will label a node by 1 if at least one subgraph in the batch contains this node. Otherwise, we will label a node by 0. We call such a labeling trick the *max-zero-one labeling trick*. In our experiments, we uniformly used the max-zero-one labeling trick for GLASS and verified its fast training speed as well as excellent empirical performance.

Previous works have explored other ways to enable batch training. For example, SEAL (Zhang & Chen, 2018) segregates small enclosing subgraphs from the whole graph to perform link prediction tasks. However, subgraphs usually contain much more nodes than links, resulting in huge enclosing subgraphs to be segregated. Moreover, in some datasets, subgraphs have multiple components. To capture the interaction among components, the depth of the enclosing subgraphs has to be large, leading to exponentially increasing subgraph sizes. IDGNN (You et al., 2021) uses fabricated additional features instead of labels, leading to loss of expressive power. Moreover, in Appendix A.6, we show that no labeling trick can avoid the inconsistencies of node labels within a batch. Therefore, it is **impossible** to solve this problem by designing a fancy labeling trick, and **approximation is a must**.

## 6 EXPERIMENT

In this section, we compare GLASS with state-of-the-art subgraph representation learning methods, especially SubGNN, on both synthetic and real-world datasets to demonstrate that GLASS is a model with superior performance and scalability, despite being much simpler than SubGNN.

### 6.1 DATASETS AND MODELS

**Datasets**. We use four synthetic datasets: `density`, `cut ratio`, `coreness`, `component`, and four real-world subgraph datasets, namely `ppi-bp`, `em-user`, `hpo-metab`, `hpo-neuro`. The four synthetic datasets are introduced by Alsentzer et al. (2020) to test a model's ability to learn the six properties: `density` tests the ability to learn internal structure; `cut ratio` tests border structure; `coreness` tests border structure and position; and `component` tests internal and external position. The four real-world datasets are also provided by Alsentzer et al. (2020). The base graph of the `ppi-bp` dataset is a human protein-protein interaction network. Each subgraph is induced by proteins in a biological process, whose label is its cellular function. The `hpo-metab` and `hpo-neuro` datasets are knowledge graphs containing phenotype and genotype information about rare diseases. Each subgraph represents a rare monogenic disease. The `em-user` dataset contains the workout history of users, where each subgraph makes up a user's workout history, and the label is the gender of the user. Detailed information on these datasets are in Appendix A.10.

**Baseline Models**. We consider four baseline methods. (1) **SubGNN** (Alsentzer et al., 2020) uses subgraph-level message passing with six artificial channels. We use the numbers provided by the original paper. (2) **Sub2Vec** (Adhikari et al., 2018) samples random walks in subgraphs which are fed to Paragraph2Vec to train subgraph embeddings. We train Sub2Vec using the official implementation to produce subgraph representations and then feed them to an MLP. Like SubGNN, Sub2Vec can also capture neighborhood and structure without passing messages between nodes. However, it uses node ID and degree ratio in the target subgraph to represent nodes, rather than pretraining GNNs. Moreover, it only learns the internal topology by sampling random walks in the target subgraph. We compare these two existing subgraph representation methods to show the superior performance of GLASS. (3) **GNN-seg** is an ordinary MPNN performing graph classification task on subgraphs segregated from the whole graph. We use it to illustrate that subgraph tasks need external topology. (4) **MLP** and (5) **GBDT** pool node embeddings to produce subgraph embed-

Table 3: Mean Micro-F1 with standard error of the mean on synthetic datasets. Results are provided from runs with ten random seeds.

| Method | density | cut ratio | coreness | component |
|--------|---------|-----------|----------|-----------|
| GLASS | $0.930 \pm 0.009$ | $\mathbf{0.935 \pm 0.006}$ | $\mathbf{0.840 \pm 0.009}$ | $\mathbf{1.000 \pm 0.000}$ |
| SubGNN | $0.919 \pm 0.006$ | $0.629 \pm 0.013$ | $0.659 \pm 0.031$ | $0.958 \pm 0.032$ |
| Sub2Vec | $0.459 \pm 0.012$ | $0.354 \pm 0.014$ | $0.360 \pm 0.019$ | $0.657 \pm 0.017$ |
| GNN-seg | $\mathbf{0.952 \pm 0.006}$ | $0.346 \pm 0.011$ | $0.593 \pm 0.012$ | $1.000 \pm 0.000$ |

Table 4: Mean Micro-F1 with standard error of the mean on real-world datasets. Results are provided from runs with ten random seeds.

| Method | ppi-bp | hpo-metab | hpo-neuro | em-user |
|--------|--------|-----------|-----------|---------|
| GLASS | $\mathbf{0.619 \pm 0.007}$ | $\mathbf{0.614 \pm 0.005}$ | $\mathbf{0.685 \pm 0.005}$ | $\mathbf{0.888 \pm 0.006}$ |
| SubGNN | $0.599 \pm 0.008$ | $0.537 \pm 0.008$ | $0.644 \pm 0.006$ | $0.816 \pm 0.013$ |
| Sub2Vec | $0.388 \pm 0.001$ | $0.472 \pm 0.010$ | $0.618 \pm 0.003$ | $0.779 \pm 0.013$ |
| GNN-seg | $0.361 \pm 0.008$ | $0.542 \pm 0.009$ | $0.647 \pm 0.001$ | $0.725 \pm 0.003$ |
| MLP | $0.445 \pm 0.003$ | $0.386 \pm 0.011$ | $0.404 \pm 0.006$ | $0.524 \pm 0.019$ |
| GBDT | $0.446 \pm 0.000$ | $0.404 \pm 0.000$ | $0.513 \pm 0.000$ | $0.694 \pm 0.000$ |

dings and classify them. They are graph-agnostic, thus their performance can show the importance of graph topology for subgraph tasks. The details of the implementation of these baselines are in Appendix A.10.

**GLASS.** For real-world datasets, we pretrain plain GNN on link prediction tasks to produce input node embeddings for GLASS. On synthetic datasets, GLASS uses homogeneous node features. We detail our GLASS architecture in Appendix A.7 and list other hyperparameters in Appendix A.10.

## 6.2 RESULTS

**Synthetic Datasets.** To evaluate the expressive power of GLASS, we test GLASS and other baselines on four synthetic datasets. Results are shown in Table 3. GLASS significantly outperforms all baselines and outperforms the previous state-of-the-art model SubGNN by $20.4\%$ on average and $48.6\%$ in maximum. The results show that GLASS can capture density and cut ratio information very well as proved. Moreover, they illustrate the expressive power of ordinary node-level MPNNs for subgraph representation learning—we do not need to use subgraph-level message passing with complicated artificial channels, but a plain GNN with a simple labeling trick is enough. As for GNN-seg, its failure to predict cut ratio and coreness is expected as these properties requires topology outside the subgraph. Similarly, Sub2Vec cannot capture position information and external topology, thus its failure on cut ratio and component is reasonable.

**Real-world datasets.**

Table 4 shows the results. GLASS gains a $8.2\%$ increase on average and $14.3\%$ increase in maximum compared with SubGNN. On `hpo-metab` and `hpo-neuro` datasets, GNN-seg beats other baselines very easily, which might be because the subgraphs in these two datasets are dense and localized, making the topology outside the subgraph less important.

In contrast, on `ppi-bp` and `em-user`, the density of subgraphs is lower, and some subgraphs are even composed of single nodes. Thus GNN-seg can learn little from the internal topology as shown by the worse performance. Sub2Vec is less expressive than SubGNN as expected. However, on `ppi-bp` and `em-user` datasets, it has an edge over GNN-seg. We suspect that the node ID can leak some external topology information to Sub2Vec, as the same nodes can appear in different subgraphs.

MLP and GBDT are both graph-agnostic models. They fail on all datasets, which illustrates that capturing graph structure is important for subgraph tasks.

**Ablation Analysis** To demonstrate the power of the max-zero-one labeling trick, we do an ablation study by removing the max-zero-one labels from GLASS, which is called GNN-plain. A comparison between GLASS and GNN-plain is shown in Table 5 (synthetic datasets) and 6 (real-world datasets). Our labeling trick improves the performance by $42.1\%$ on synthetic datasets and $2.8\%$ on real-world

Table 5: Mean Micro-F1 with standard error of the mean on synthetic datasets. Results are provided from runs with ten random seeds.

| Method | density | cut ratio | coreness | component |
|---|---|---|---|---|
| GLASS | $\mathbf{0.930 \pm 0.009}$ | $\mathbf{0.935 \pm 0.006}$ | $\mathbf{0.840 \pm 0.009}$ | $\mathbf{1.000 \pm 0.000}$ |
| GNN-plain | $0.462 \pm 0.011$ | $0.887 \pm 0.008$ | $0.536 \pm 0.008$ | $0.998 \pm 0.002$ |

Table 6: Mean Micro-F1 with standard error of the mean on real-world datasets. Results are provided from runs with ten random seeds.

| Method | ppi-bp | hpo-metab | hpo-neuro | em-user |
|---|---|---|---|---|
| GLASS | $\mathbf{0.619 \pm 0.007}$ | $\mathbf{0.614 \pm 0.005}$ | $\mathbf{0.685 \pm 0.005}$ | $\mathbf{0.888 \pm 0.006}$ |
| GNN-plain | $0.613 \pm 0.009$ | $0.597 \pm 0.012$ | $0.668 \pm 0.007$ | $0.847 \pm 0.021$ |

datasets on average. The relatively lower improvement on real-world datasets might be because improving the expressive power is not the most critical factor for real-world datasets, but rather, filtering noise and smoothing features is more effective. In other words, the synthetic datasets require a much stronger structure learning ability than the real-world datasets to perform well. This can also be seen from the comparison between SubGNN and GNN-seg in Table 4, where SubGNN only marginally outperforms GNN-seg on real-world datasets.

In Appendix A.11, we analyze the effect of batch size for GLASS on some datasets. In general, the performance of GLASS drops as the batch size increases. However, the max-zero-one labeling trick can still boost plain GNNs at a large batch size. The performance gap between GLASS and GNN also shows that the effect of overlapping node labels is insignificant under our experimental setting. In Appendix A.9, we also design a score to quantify the overlapping effect.

**Computation Time Comparison** The precomputation of SubGNN is quite time-consuming. After precomputation, the training of SubGNN is also slower than GLASS. On the 7 used datasets (except `ppi-bp`), SubGNN on average takes $2.7\times$ more time than GLASS to converge. Moreover, the ratio jumps to 7.9 when we count in precomputation. The results can be found in Figure 3.

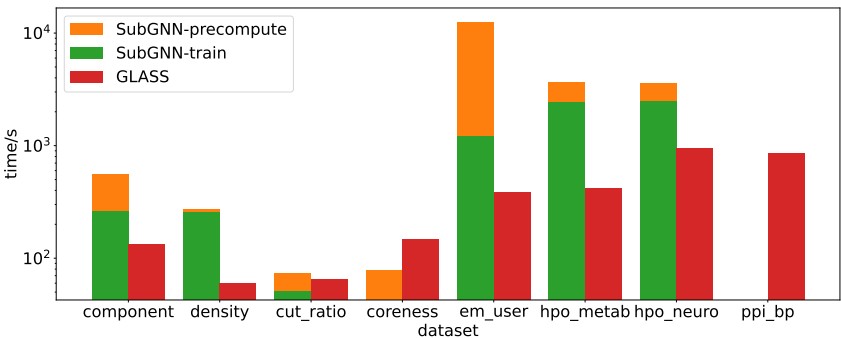

Figure 3: The time needed for training a model (log scale). SubGNN takes more than 48h on `ppi-bp`.

## 7 CONCLUSION

We have proposed GLASS, a simple yet powerful model for subgraph representation learning. We illustrate the failure of plain GNNs on some simplest subgraph learning tasks, and demonstrate the power of labeling trick to enhance plain GNNs both theoretically and empirically. GLASS outperforms the previous state-of-the-art method SubGNN by $20.4\%$ on synthetic datasets and $8.2\%$ on real-world datasets, verifying the power of node-level message passing for subgraph representation learning. We argue that plain GNNs + labeling trick are enough for learning higher-order substructures, and the complicated subgraph-level message passing is not necessary. Despite the success of our max-zero-one labeling trick, it is still less powerful and heuristic compared with other existing labeling tricks. We leave the exploration of other labeling tricks for subgraph tasks to future work.

## 8 REPRODUCIBILITY STATEMENT

Our main theoretical contribution is in Appendix A.5, and complete proofs are also shown there. Moreover, our proposed model GLASS consists of our max-zero-one labeling trick and a plain GNN. We describe the labeling trick in Section 5.2 and the architecture of GLASS in Appendix A.7 in detail. All datasets used in our experiments are public. Our code is available at `https://github.com/Xi-yuanWang/GLASS`.

## ACKNOWLEDGEMENTS

The authors greatly thank the actionable suggestions from the reviewers. Zhang is partly supported by the CCF-Baidu Open Fund (NO.2021PP15002000).

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

## A APPENDIX

### A.1 OTHER STRUCTURAL ENCODING METHODS

We discuss other existing structural encoding methods as follows.

**Positional Encoding in Graph Transformers.** WL-PE (Zhang et al., 2020) utilizes the Weisfeiler-Lehman algorithm to produce node labels. As GNN can be as expressive as the WL test, this technique in theory will not increase the expressive power of GNNs. From the design of labeling tricks, Laplacian-PE (Dwivedi & Bresson, 2020) is not permutation invariant and cannot differentiate nodes within and outside the target subgraph, as isomorphic nodes can have different labels, while non-isomorphic nodes can have the same label (for example the eigenvector corresponding to eigenvalue 0). Graphormer proposes centrality encoding, which encodes the indegree and outdegree of nodes to help Transformer capture graph structure information. (Ying et al., 2021a). Though it is specifically useful for graph transformers, GIN (Xu et al., 2019) can easily capture node degree with one message passing layer.

**Edge and Path Feature.** They are not designed for subgraph tasks and cannot represent node position relative to the target subgraph. Spagan (Yang et al., 2019), Graphormer (Ying et al., 2021a), and PAGTN (Chen et al., 2019) both utilize shortest path encodings to reconstruct the adjacency matrix. They can capture long-range node interaction, which is important for subgraph tasks, but orthogonal to our zero-one labeling trick to distinguish nodes inside and outside the target subgraph. And we can easily add our labeling trick to EGNN (Gong & Cheng, 2019), GINE+ (Brossard et al., 2020).

### A.2 PROOF OF THEOREM 1

*Proof.* Consider the following GNN, which ignores node features.

$$a_v^{(1)} = \sum_{u \in N(v)} \left( l_u^{(\mathcal{S})} \begin{bmatrix} 1 \\ 0 \\ 0 \end{bmatrix} + \begin{bmatrix} 0 \\ 1 \\ 0 \end{bmatrix} \right) \tag{7}$$

$$h_v^{(1)} = \begin{bmatrix} 1 & 0 & 0 \\ -1 & 1 & 0 \\ 0 & 0 & 0 \end{bmatrix} a_v^{(1)} + \begin{bmatrix} 0 \\ 0 \\ 1 \end{bmatrix} \tag{8}$$

$$h_{\mathcal{S}} = \sum_{v \in \mathbb{V}_{\mathcal{S}}} h_v^{(1)}, \tag{9}$$

where $l_v^{(\mathcal{S})}$ is the zero-one label of node $v$, $h_{\mathcal{S}}$ is the representation of subgraph $\mathcal{S}$.

Density $d$ and cut ratio $c$ can be predicted with GLASS in a transductive setting.

$$d(\boldsymbol{h}_{\mathcal{S}}) = (\begin{bmatrix} 1 \\ 0 \\ 0 \end{bmatrix}^T \boldsymbol{h}_{\mathcal{S}})/[(\begin{bmatrix} 0 \\ 0 \\ 1 \end{bmatrix}^T \boldsymbol{h}_{\mathcal{S}}) \cdot (\begin{bmatrix} 0 \\ 0 \\ 1 \end{bmatrix}^T \boldsymbol{h}_{\mathcal{S}} - 1)] \tag{10}$$

$$c(\boldsymbol{h}_{\mathcal{S}}) = (\begin{bmatrix} 0 \\ 0.5 \\ 0 \end{bmatrix}^T \boldsymbol{h}_{\mathcal{S}})/[(\begin{bmatrix} 0 \\ 0 \\ 1 \end{bmatrix}^T \boldsymbol{h}_{\mathcal{S}}) \cdot (n - \begin{bmatrix} 0 \\ 0 \\ 1 \end{bmatrix}^T \boldsymbol{h}_{\mathcal{S}} - 1)]. \tag{11}$$

$\square$

### A.3  PROOF OF PROPOSITION 1

*Proof.* First, we need to prove that given any plain GNN model $m_1$, there exists a GLASS model $m_2$ producing the same output for any target subgraph $\mathcal{S}$.

Assuming that the AGGREGATE function of $m_1$ is $f_1^{(k)}$, the COMBINE function of $m_1$ is $g_1^{(k)}$ at the $k^{\text{th}}$ layer and the READOUT function of $m_1$ is $\phi_1$. We can define a function $\theta$ mapping labeled node features to the initial node features, $\theta(\text{CONCATENATE}(\boldsymbol{h}_u^{(k-1)}, \boldsymbol{l}^{(\mathcal{S})})) = \boldsymbol{h}_u^{(k-1)}$, which is a well defined function. And we can design a GLASS as follows.

$$\boldsymbol{h}_u'^{(k-1)} = \text{CONCATENATE}(\boldsymbol{h}_u^{(k-1)}, \boldsymbol{l}^{(\mathcal{S})}), \tag{12}$$

$$\boldsymbol{a}_v^{(k)} = f_1^{(k)}(\{\theta(\boldsymbol{h'}_u^{(k-1)})|u \in N(v)\}), \tag{13}$$

$$\boldsymbol{h}_v^{(k)} = g_1^{(k)}(\boldsymbol{h}_v^{(k-1)}, \boldsymbol{a}_v^{(k)}), \tag{14}$$

where $\boldsymbol{h}_v^{(k)}$ is the embedding of node $v$ at the $k^{\text{th}}$ layer.

We use $\phi_1$ as the READOUT function, therefore $m_2$ will produce the same embeddings as $m_1$ for the target subgraph.

Second, GLASS can differentiate at least one pair of subgraphs which plain GNNs cannot distinguish. Figure 2 gives us an example. $\square$

### A.4  ANALYSIS OF HOW GLASS CAN COVER THE SIX PROPERTIES IN SUBGNN

The implementation of the channels in SubGNN are non-deterministic, so GLASS cannot precisely reproduce the behavior of SubGNN. Our goal is to prove that GLASS can represent the six channels by definition.

As the SubGNN uses embeddings produced by plain GNNs as node features, we first show that GLASS can also produce such node embeddings.

**Proposition 3.** *Given a graph $\mathcal{G}$ and node embeddings $\boldsymbol{G}$ produced by plain GNNs, GLASS can also produce the same embeddings.*

*Proof.* We have prove that GLASS is more expressive than plain GNNs in 1. $\square$

Therefore, for any GLASS model, we can use some bottom layers to produce node embedding, and the rest of the model is still a GLASS. Therefore, we can assume the input node features of GLASS are the same as those of SubGNN. Let $\boldsymbol{g}_u$ denote the pretrained GNN embeddings of node $u$.

We also need to show the performance of plain GNNs. Xu et al. (2019) prove that some plain GNNs map two nodes to the same embeddings only if they have identical BFS structures with identical features on the corresponding nodes. Assuming the $k$-depth rooted subtree set is $\mathbb{T} = \{\mathcal{T}_1, \mathcal{T}_2, ..., \mathcal{T}_{|\mathbb{V}|}\}$, where $\mathcal{T}_u$ means the tree rooted in node $u$. If READOUT is an injective multiset function like DeepSet (Zaheer et al., 2017), plain GNN will be an injective function $\psi : \{\mathbb{T}_{\mathcal{S}}|\mathcal{S} \subseteq \mathcal{G}\} \rightarrow \mathbb{R}^d$, where $\mathbb{T}_{\mathcal{S}} = \{\mathcal{T}_u|u \in \mathbb{V}_{\mathcal{S}}\}$.

The internal N property encodes the set of nodes in the target subgraph, and the border N property encodes the set of neighbors within $k$ hops of nodes in the target subgraph. Therefore, to show that GLASS can cover the N channel in SubGNN, we only need to prove that GLASS can differentiate any pair of neighbor feature sets. See Proposition 4.

**Proposition 4.** *Given a target subgraph $\mathcal{S}$ in a graph $\mathcal{G}$, GLASS can produce different embeddings for any pair of different border neighbor feature sets $\{\boldsymbol{g}_u | d(u,v) \leq k, v \in \mathbb{V}_\mathcal{S}\}$ or internal neighbor feature sets $\{\boldsymbol{g}_v | v \in \mathbb{V}_\mathcal{S}\}$.*

*Proof.* For border neighbor feature sets, let function $f(\mathbb{T}_\mathcal{S}) = \{\boldsymbol{g}_u | u \in \mathbb{V}_\mathcal{T}, \mathcal{T} \in \mathbb{T}_\mathcal{S}\}$. $\psi^{-1} \circ f$ is a function mapping embeddings to border neighbor set. Therefore, GLASS can produce different embeddings for any pair of different border neighbor feature sets $\{\boldsymbol{g}_u | d(u,v) \leq k, v \in \mathbb{V}_\mathcal{S}\}$.

As for the internal neighbor feature set, we can use a GLASS layer whose COMBINE function maps all nodes labeled zero to a constant embedding vector $\boldsymbol{g}_0$ different from any existing initial node embeddings, and maintain the input embeddings of nodes labeled one and pass it to a GLASS used in last paragraph. Let function $\theta(\mathbb{U}) = \mathbb{U} - \{\boldsymbol{g}_0\}$. Therefore, $\psi^{-1} \circ f \circ \theta$ maps embeddings to internal neighbor feature sets. Therefore, GLASS can produce different embeddings for any pair of different internal neighbor feature sets $\{\boldsymbol{g}_v | v \in \mathbb{V}_\mathcal{S}\}$. $\square$

For the P channel, the internal position is defined as the distance (shortest path length) between nodes in the target subgraph, and the border position is the distance between nodes in the target subgraph and nodes in the whole graph.

**Proposition 5.** *Given a target subgraph $\mathcal{S}$ in graph $\mathcal{G}$, if embeddings of nodes are different from each other, GLASS can produce different embeddings for subgraphs with different position information.*

*Proof.* $k$-layer MPNN can capture rooted subtree structure. As the node embeddings are different, the set of embeddings of nodes can be mapped to the set of nodes bijectively. If $\boldsymbol{g}_u$ appear in $l$-depth rooted subtree but not in $l-1$-depth rooted subtree, we can infer that the distance between the target node and node $u$ is $l$. Moreover, using the DeepSet function as READOUT, we can capture the distance between nodes in the target subgraph and any other nodes. $\square$

Note that though we assume that the node embeddings are different from each other in Proposition 5, we can let the input feature be learnable in implementation. Thus the coincidence of node embeddings can be avoided.

Moreover, with such technique, we can build a bijective function from the embedding space to the node set. The theoretical expressive power of GLASS will be stronger and no longer be limited to node embedding generated by plain GNNs.

As for the S channel, it is defined as the internal feature connectivity (the multiset $\{(\boldsymbol{g}_u, \boldsymbol{g}_v) | (u,v) \in \mathbb{E}_\mathcal{S}\}$), and the border feature connectivity (the multiset $\{(\boldsymbol{g}_u, \boldsymbol{g}_v) | u \in \mathbb{V}_\mathcal{S}, v \notin \mathbb{V}_\mathcal{S}, (u,v) \in \mathbb{E}\}$).

**Proposition 6.** *GLASS can represent the internal and border structure. In other words, there exists a GLASS model mapping the multiset $\{(\boldsymbol{g}_u, \boldsymbol{g}_v) | (u,v) \in \mathbb{E}_\mathcal{S}\}$ and $\{(\boldsymbol{g}_u, \boldsymbol{g}_v) | u \in \mathcal{S} \text{ and } v \notin \mathcal{S}\}$ to the embedding space injectively, where $\boldsymbol{g}$ is the pretrained GNN embeddings.*

*Proof.* WLGNN can encode the multiset of feature pairs $\{\{(\boldsymbol{g}_u, \boldsymbol{g}_v) | v \in N(u)\} | u \in \mathbb{V}_\mathcal{S}\}$ with the following encoder.

$$\sum_{u \in \mathbb{V}_\mathcal{S}} f_1(\sum_{v \in N(u)} f_2(\boldsymbol{g}_v)), \tag{15}$$

where $f_1$ and $f_2$ are some functions whose existence Xu et al. (2019) have proved. In other words, we can encode BFS trees whose depth=1.

If we concatenate the node labels and $\boldsymbol{g}$ (namely $\boldsymbol{g}'$), and design

$$f_2' = \begin{cases} f_2(\boldsymbol{g}_v) & \text{if } L_v = 1 \\ 0 & \text{otherwise} \end{cases} \tag{16}$$

$$f_2'' = \begin{cases} f_2(\boldsymbol{g}_v) & \text{if } L_v = 0 \\ 0 & \text{otherwise} \end{cases} \tag{17}$$

$\sum_{u \in \mathbb{V}_{\mathcal{S}^{(c)}}} f_1'(\sum_{v \in N(u)} f_2(\boldsymbol{g}_v))$ can represent $\{(\boldsymbol{g}_u, \boldsymbol{g}_v) | (u, v) \in \mathbb{E}_{\mathcal{S}^{(c)}}\}$.

And $\sum_{u \in \mathbb{V}_{\mathcal{S}^{(c)}}} f_1''(\sum_{v \in N(u)} f_2(\boldsymbol{g}_v))$ can represent $\{(\boldsymbol{g}_u, \boldsymbol{g}_v) | u \in \mathcal{S}^{(c)} \text{ and } v \notin \mathcal{S}^{(c)}\}$.  $\square$

## A.5 THE EXPRESSIVE POWER OF GLASS UNDER STRUCTURAL REPRESENTATION THEORY

Zhang et al. (2021) introduce a theoretical framework to analyze the expressive power of GNNs. Using this framework, we theoretically characterize the expressive power of GLASS. We first introduce some terms of labeling trick theory.

A *permutation* $\pi$ is a bijective mapping from $\{1, 2, ..., n\}$ to $\{1, 2, ..., n\}$, where $n \in \mathbb{N}^+$. All possible permutations constitute the permutation group $\Pi_n$. For node set $\mathbb{U} \subseteq \mathbb{V}$, $\pi(\mathbb{U}) = \{\pi(i) | i \in \mathbb{U}\}$, where $i$ is the node index in a graph $\mathcal{G}$, and $\pi(i)$ can be node index in $\mathcal{G}$ or any other graph. For edge set $\mathbb{F} \subseteq \mathbb{E}$, $\pi(\mathbb{F}) = \{(\pi(i), \pi(j)) | (i, j) \in \mathbb{F}\}$. For node attribute matrix $\boldsymbol{X}$, $\pi(\boldsymbol{X})_{\pi(i)} = \boldsymbol{X}_i$. And for subgraph $\mathcal{S}$ and graph $\mathcal{G}$, $\pi(\mathcal{S}) = (\pi(\mathbb{V}_S), \pi(\mathbb{E}_S), \pi(X_S))$ and $\pi(\mathcal{G}) = (\pi(\mathbb{V}), \pi(\mathbb{E}), \pi(\boldsymbol{X}))$.

The definition of labeling trick is as follows.

**Definition 2.** *Given a subgraph $\mathcal{S}$ in a graph $\mathcal{G}$, we design a label matrix $\boldsymbol{L}^{(\mathcal{S})} \in \mathbb{R}^{|\mathbb{V}| \times d}$, whose $u^{th}$ is the label of node u. $\boldsymbol{L}$ satisfies: for any two subgraphs $\mathcal{S}$ and $\mathcal{S}'$ in $\mathcal{G}$*

*1. (target-nodes-distinguishing) $\boldsymbol{L}^{(\mathcal{S})} = \pi(\boldsymbol{L}^{(\mathcal{S}')}) \Rightarrow \mathbb{V}_{\mathcal{S}} = \pi(\mathbb{V}_{\mathcal{S}'})$*

*2. (permutation equivariance) $\mathcal{S} = \pi(\mathcal{S}'), \mathcal{G} = \pi(\mathcal{G}) \Rightarrow \boldsymbol{L}^{(\mathcal{S})} = \pi(\boldsymbol{L}^{(\mathcal{S}')})$.*

Proved by Zhang et al. (2021), the zero-one labeling trick is a valid labeling trick.

Now we analyze the power of labeling trick. We first define *subgraph isomorphism*.

**Definition 3.** *Given two n-node graphs $\mathcal{G} = (\mathbb{V}, \mathbb{E}, \boldsymbol{X})$, $\mathcal{G}' = (\mathbb{V}', \mathbb{E}', \boldsymbol{X}')$, and one subgraph $\mathcal{S}$ of $\mathcal{G}$ and one subgraph $\mathcal{S}'$ of $\mathcal{G}'$, $\mathcal{S}$ and $\mathcal{S}'$ are isomorphic (denoted by $\mathcal{S} \simeq \mathcal{S}'$) iff $\exists \pi \in \Pi_n$, $\pi(\mathcal{S}) = \pi(\mathcal{S}')$ and $\pi(\mathcal{G}) = \pi(\mathcal{G}')$.*

Isomorphic subgraphs should be mapped to the same embeddings, and the most expressive representations should differentiate all non-isomorphic subgraphs. Therefore we introduce the concept of *structural subgraph embeddings* to illustrate such perfect representations.

**Definition 4.** *$\Gamma(\mathcal{S})$ is a **structural subgraph embedding** for subgraph $\mathcal{S}$ if $\forall \mathcal{S}, \mathcal{S}', \Gamma(\mathcal{S}) = \Gamma(\mathcal{S}') \Leftrightarrow \mathcal{S} \simeq \mathcal{S}'$*

Numerous works have been exploring the expressive power of GNNs on node tasks. However, we focus on how labeling trick can help GNNs on subgraph representation learning. Therefore, we introduce an ideal GNN model for node tasks.

**Definition 5.** *If a GNN can map a node u to the embedding vector $\boldsymbol{h}_u$ so that given any two graphs $\mathcal{G}$ and $\mathcal{G}'$, $i \in \mathbb{V}, j \in \mathbb{V}'$, $\boldsymbol{h}_i = \boldsymbol{h}_j \Leftrightarrow \exists \pi \in \Pi_n, i = \pi(j)$ and $\mathcal{G} = \pi(\mathcal{G}')$, we call it a node-most-expressive GNN.*

Though GNN is powerful on various tasks, Zhang et al. (2021) show that node embeddings generated by GNNs are not enough to produce structural subgraph embedding even if we use a node-most-expressive GNN. However, labeling trick can solve this problem. Zhang et al. (2021) prove that with labeling trick and injective READOUT function, node-most-expressive GNNs can produce structural subgraph embeddings.

**Theorem 2.** *There exists function $\Gamma$, where $\Gamma(\mathcal{S})$ is structural subgraph embedding for subgraph $\mathcal{S}$, can be formulated as $\Gamma(\mathcal{S}) = READOUT(\{H_v | v \in \mathbb{V}_S\})$, where $\boldsymbol{H}$ is the node embedding matrix of a node-most-expressive GNN with node label as input.*

Please refer to Appendix A of (Zhang et al., 2021) for the proof.

However, up to now, no scalable node-most-expressive GNNs have been realized. However, with less expressive GNNs, labeling trick can still make a difference. Figure 2 provides an example.

Here we prove that labeling trick can boost WLGNNs, a kind of GNNs whose expressive power is the same as 1-WL.

**Theorem 3.** *For any $\epsilon \in \mathbb{R}^+$, given any graph $\mathcal{G}$ with n nodes, whose node degrees range from 1 to $O(\log^{(1-\epsilon)/2} n)$, and having an empty feature, there exists $w(2^n n^{2\epsilon-1})$ pairs of non-isomorphic subgraphs such that any h-layer WLGNN produces the same representation, while with labeling trick WLGNN can distinguish them.*

*Proof.* We prove that there are $w(n^{2\epsilon})$ nodes that WLGNNs cannot differentiate in (1), and get the bound in (2).

(1)The bound of the number of nodes in a n-hop BFS tree is

$$|V(G_v^{(h)})| \leq K = \sum_{i=0}^{h} d^i = \mathcal{O}(d^h) = \mathcal{O}(\log^{\frac{1-\epsilon}{2}} n). \tag{18}$$

We can add virtual nodes to make the trees become complete k-ary trees. And by enumerating the position of the virtual node we can get a bound $2^{C_K^2} = O(n^{1-\epsilon})$ for the number of non-isomorphic trees.

According to the pigeonhole principle, there exist $n/\mathcal{O}(n^{1-\epsilon}) = n^\epsilon$ nodes which cannot be differentiated with each other by WLGNNs. They form a set $\mathbb{V}_{iso}$.

(2) Let us partition $\mathbb{V}_{iso} = \bigcup_{i=1}^{q} \mathbb{V}_i$, nodes in each $V_i$ share the same one-hop neighbor. Consider a node $u \in \mathbb{V}_i, v \in \mathbb{V}_j, i \neq j$. There exists a node $w \in N(u), w \notin N(v)$. Let $\tilde{\mathbb{V}}_v$ donate $\mathbb{V} - \{u, v, w\} - N(w)$. $|\mathbb{V}_v| \geq w(n - \log^{\frac{1-\epsilon}{2h}} n)$. Consider arbitrary subset $\tilde{\mathbb{V}}_v \subseteq \mathbb{V}$. Let $\mathcal{S}_1$ donate the subgraph induced by $\tilde{\mathbb{V}}_v \bigcup \{u, w\}$, $\mathcal{S}_2$ donate the subgraph induced by $\tilde{\mathbb{V}}_v \bigcup \{v, w\}$. The density of $\mathcal{S}_1$ is higher than $\mathcal{S}_2$. And GLASS can fit density perfectly, so GLASS can distinguish $\mathcal{S}_1$ and $\mathcal{S}_2$, while WLGNNs cannot.

And the number of pairs $(u, v, w)$ is

$$\prod_{i,j=1,i\neq j}^{q} |\mathbb{V}_i||\mathbb{V}_j| = \frac{1}{2}(|\mathbb{V}_{iso}| - \sum_{i=1}^{q} |V_i|^2). \tag{19}$$

As nodes in $|V_i|$ share the same neighbor, $|V_i| \leq O(\log^{\frac{1-\epsilon}{2h}} n)$. So the expression above

$$\prod_{i,j=1,i\neq j}^{q} |\mathbb{V}_i||\mathbb{V}_j| = w(n^{2\epsilon}) \tag{20}$$

And the bound for the number of these pairs of subgraphs is

$$w(n^{2\epsilon}) 2^{w(n-\log^{\frac{1-\epsilon}{2h}} n)} = w(2^n n^{2\epsilon-1}). \tag{21}$$

$\square$

## A.6 THEORETICAL ANALYSIS OF MAX-ZERO-ONE LABELING TRICK

Take a closer look at the definition of labeling trick. Node label should be (1) target-nodes-distinguishing. Labels of nodes in $\mathcal{S}$ must be different from those out of $\mathcal{S}$. (2) permutation equivariance. Isomorphic nodes (in $\mathcal{S}$ and out of $\mathcal{S}$ separately) must be mapped to the same label. We can illustrate the function of the conditions with the following theorem.

**Theorem 4.** *For any $\mathcal{S}, \mathcal{S}'$ in given graphs $\mathcal{G}, \mathcal{G}'$ respectively, there exists a node-most-expressive GNN with labeling trick satisfying: (1) $\boldsymbol{h}_\mathcal{S} = \boldsymbol{h}_{\mathcal{S}'} \Rightarrow \mathcal{S} \simeq \mathcal{S}'$ if $\boldsymbol{l}^{(\mathcal{S})} = \pi(\boldsymbol{l}^{(\mathcal{S}')}) \Rightarrow \mathbb{V}_\mathcal{S} = \pi(\mathbb{V}_{\mathcal{S}'})$; (2) $\boldsymbol{h}_\mathcal{S} = \boldsymbol{h}_{\mathcal{S}'} \Leftarrow \mathcal{S} \simeq \mathcal{S}'$ if $\mathcal{S} = \pi(\mathcal{S}'), \mathcal{G} = \pi(\mathcal{G}) \Rightarrow \boldsymbol{l}^{(\mathcal{S})} = \pi(\boldsymbol{l}^{(\mathcal{S}')})$, where $\boldsymbol{h}$ is GNN embedding of subgraph.*

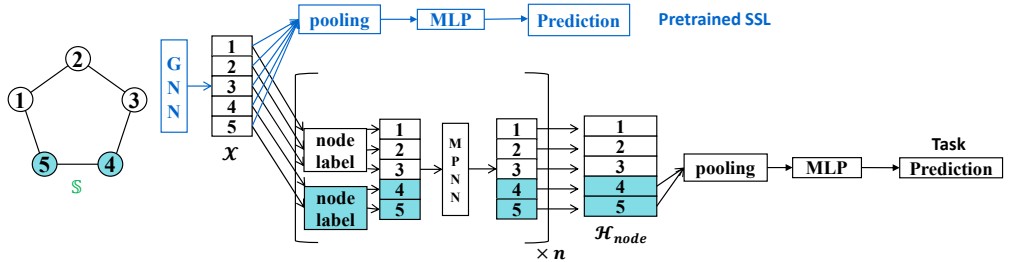

Figure 4: GLASS is composed of node label function, MPNN, pooling layers.

This theorem has been proven in Appendix A of (Zhang et al., 2021).

**Proposition 7.** *For all labeling tricks, there exists a case that given two different subgraphs $\mathcal{S}$, $\mathcal{S}'$ in graph $\mathcal{G}$, node label $\boldsymbol{l}^{(\mathcal{S})} \neq \boldsymbol{l}^{(\mathcal{S}')}$.*

*Proof.* Consider three isomorphic nodes, namely $u, v, w$, in a graph. Let $\mathcal{S}_1$ donate subgraph induced by $\{u, v\}$, $\mathcal{S}_2$ donate subgraph induced by $\{u, w\}$. Therefore, $L_u^{(\mathcal{S}_1)} = L_v^{(\mathcal{S}_1)} \neq L_w^{(\mathcal{S}_1)}$ and $L_u^{(\mathcal{S}_2)} = L_w^{(\mathcal{S}_2)} \neq L_v^{(\mathcal{S}_2)}$. $L^{(\mathcal{S}_1)} \neq L^{(\mathcal{S}_2)}$. □

Therefore, developing less theoretically powerful tricks is a must. Here, we introduce the max-zero-one labeling trick. Looking back on the definition of labeling tricks, it is invariant but not expressive enough.

**Definition 6.** *Given subgraphs in a batch $\{\mathcal{S}_1, ..., \mathcal{S}_n\}$. The max-zero-one label of a node $v$ in $\mathcal{G}$ is*

$$L_v = \begin{cases} 1 & if \ \exists j \in [1, 2, ..., n], v \in \mathbb{V}_{\mathcal{S}_j} \\ 0 & otherwise \end{cases} \tag{22}$$

With Theorem 4, we can prove the invariance of the max-zero-one node labels.

**Corollary A.1.** *With a node-most-expressive GNN, injective READOUT and the max-zero-one node labels, in a single batch, $\mathcal{S} \simeq \mathcal{S}' \Rightarrow \boldsymbol{H}_{\mathcal{S}} = \boldsymbol{H}_{\mathcal{S}'}$*

Though the max-zero-one results in loss of expressive power compared with zero-one, experiments show that the model can fit cut ratio and density very well. These two properties need a clear division of nodes in and outside the target subgraph.

## A.7 GLASS ARCHITECTURE

GLASS takes embeddings produced by pretrained GNN as input node feature. Then it uses node label function to transform node embeddings before message passing at each label, where node label functions are node label-specific MLPs. Then GLASS pools node embeddings to produce subgraph embeddings and use a mlp to produce the output. We also use normalization layers (Cai et al., 2021) to accelerate optimization.

## A.8 SELF-SUPERVISED LEARNING

SubGNN uses GNN pretrained on link prediction task to produce input node embeddings for GLASS. We also use it to compare GLASS and GNN-plain with SubGNN. Pretrained GNN embeddings can help GLASS capture distant neighbors. If the pretrained GNN has $l$ layers and the GNN used in GLASS has $k$ layers, it can capture $(l + k)$-hop neighbors, while a $k$-layer MPNN can only capture neighbors within $k$ hops. Moreover, we design more SSL tasks to further boost GLASS.

In practice, subgraph datasets are often not large, and target subgraphs are often sparse in the whole graph. Thus, training GLASS only from the subgraph property prediction signals tends to overfit. Therefore, we introduce three levels of self-supervised learning (SSL) tasks to assist the GLASS

Table 7: Mean Micro-F1 with standard error of the mean of GLASS with SSL on test sets. esults are provided from runs with ten random seeds.

| dataset | GLASS | GLASS+SSL |
|---|---|---|
| em_user | $0.888 \pm 0.006$ | $\mathbf{0.902 \pm 0.006}$ |
| ppi_bp | $0.619 \pm 0.007$ | $\mathbf{0.621 \pm 0.008}$ |
| hpo_metab | $\mathbf{0.614 \pm 0.005}$ | $0.565 \pm 0.006$ |
| hpo_neuro | $\mathbf{0.685 \pm 0.005}$ | $0.674 \pm 0.002$ |

Table 8: Noisy label ratio for each datasets under our experimental setting.

| dataset | $nr_1$ | $nr_2$ | dataset | $nr_1$ | $nr_2$ |
|---|---|---|---|---|---|
| density | 0.022 | 0.012 | em_user | 0.018 | 0.011 |
| cut_ratio | 0.018 | 0.012 | ppi_bp | 0.078 | 0.044 |
| coreness | 0.023 | 0.024 | hpo_metab | 0.063 | 0.040 |
| component | 0.004 | 0.028 | hpo_neuro | 0.097 | 0.061 |

training. 1) Node level: We train a GNN to produce node embeddings to predict $\boldsymbol{A}^k \boldsymbol{X}$, where $\boldsymbol{A}$ is the adjacency matrix of the whole graph. Intuitively, it helps GNNs differentiate different BFS trees (Yehudai et al., 2021), thus preserving better local subtree information. 2) Edge level: We perform a link prediction task on the whole graph. Intuitively, it pushes nearby nodes to have similar embeddings, thus encoding the node distance information into the node embeddings. 3) Subgraph level: We let the final subgraph representations predict some subgraph structural properties, including density, cut ratio, coreness, and the number of connected components. It helps the model better capture some higher-order substructure information.

We take a pretraining strategy for node and edge-level SSL tasks, whose output node embeddings are used as initial features of the GLASS. Then, we jointly train the subgraph-level SSL tasks and the main subgraph property prediction task. Donate the loss for the main task, node-level SSL, edge-level SSL, and subgraph-level SSL as $L_{main}, L_{node}, L_{edge}, L_{subg}$. The pretraining loss is given by $L_{pre} = \alpha L_{node} + \beta L_{edge}$, and the joint training loss is given by $L_{joint} = L_{main} + \gamma L_{subg}$, where $\alpha, \beta, \gamma$ are hyperparameters.

We only use SSL for real-world datasets. The results are listed in Table 7.

SSL tasks boost GLASS on em_user and ppi_bp datasets.

## A.9 NOISY LABEL RATIO

In this paper, we only analyse the expressive power of zero-one labeling trick but use max-zero-one labeling trick in experiments. If subgraphs are so sparsely located in the subgraph that their $k$-hop neighbors do not overlap, both tricks are equivalent for a $k$-layer GLASS. However, many real-world graphs have a small diameter, so such overlap maybe inevitable. To quantify the overlapping effect, we define *noisy label ratio* $nr_i$: the ratio of $i$-hop neighbors with inconsistent zero-one and max-zero-one labels computed over all subgraph nodes. In Table 8, we show the nnoisy label ratio of the datasets under our experimental settings. The results show that the overlapping effect brought by using max-zero-one labeling trick is insignificant.

## A.10 IMPLEMENTATION DETAILS

**Datasets.** We use the code provided by SubGNN to produce synthetic datasets and use the real-world datasets provided by SubGNN directly. The statistics of these datasets can be found in Table 9. As for dataset division, the real-world datasets take an 80:10:10 split, and the synthetic datasets follow a 50:25:25 split, following (Alsentzer et al., 2020).

**Computing infrastructure.** We leverage Pytorch Geometric and Pytorch for model development. Models were trained on an Nvidia V100 GPU to measure the train time and were tested on an Nvidia A40 GPU on a Linux server.

**Implementation of Message Passing Networks.** For all message-passing network models, we try three kinds of aggregation methods: sum, mean, and GCN and four kinds of pooling methods: sum,

Table 9: Detail of datasets

| Datasets | $|\mathbb{V}|$ | $|\mathbb{E}|$ | Number of Subgraphs |
|---|---|---|---|
| density | 5,000 | 29,521 | 250 |
| cut-ratio | 5,000 | 83,969 | 250 |
| coreness | 5,000 | 118,785 | 221 |
| component | 19,555 | 43,701 | 250 |
| ppi_bp | 17,080 | 316,951 | 1,591 |
| hpo_metab | 14,587 | 3,238,174 | 2,400 |
| hpo_neuro | 14,587 | 3,238,174 | 4,000 |
| em_user | 57,333 | 4,573,417 | 324 |

mean, max, and size, where the size pooling is to sum the node representations and divide it by the square root of the number of nodes.

**Input node feature.** We use an all-one vector for GNN-plain, GNN-seg, Graphormer-seg, and GLASS on synthetic datasets. On real-world datasets, GLASS, GNN-plain, MLP, and GBDT use trainable pretrained node embedding. GNN-seg uses node degree.

**Pretraining GNN.** Pretrained node embeddings were pretrained using a $l$-layer GNN with node id as the input node feature. Fixed hyperparameters were batch size $= 131072$, learning rate $= 1e - 3$, hidden dimension $= 64$. Dropout is selected from $[0.0, 0.5]$ and $l$ ranges from 1 to 5.

**Baselines.** We directly use the results reported for SubGNN. Sub2Vec is designed for connected subgraphs, so we sample random walks at each component separately. We utilize the implementation of Sub2Vec and use all its channels. For MLP and GBDT, We first pool the node embeddings to produce subgraph embeddings, then use MLP and GBDT (implemented with xgboost) to classify subgraph embeddings.

**Model hyperparameter tuning.** We use optuna to perform random search. Hyperparameters were selected to optimize Micro-F1 scores on the validation sets. The best hyperparameters selected for each model can be found in our code in the supplement materials. For GLASS, we select the learning rate from $\{1e - 4, 2e - 4, 5e - 4, 1e - 3, 2e - 3, 5e - 3\}$; number of layers from $\{1, 2\}$; hidden dimension, 64 for real-world datasets and $\{1, 5, 9, 13, 17\}$; dropout, 0.5 for real-world datasets and $\{0.1, 0.2, 0.3\}$ for synthetic datasets; aggregation, $\{$mean, sum, gcn$\}$; pool, $\{$mean, sum, max, size$\}$; batch size, $\{ns/80, ns/40, ns/20, ns/10\}$, where $ns$ is the size of datasets.

**Training process.** We set an upper bound (10000) for the number of forward and backward processes and use an early stop strategy which finishes training if the validation score does not increase after 1000 forward and backward processes. We utilize Adam optimizer and ReduceLROnPlateau learning rate scheduler to optimize models.

## A.11 Trade off among batch size, training time, and performance

On coreness, cut ratio, and density datasets, we test the performance of GLASS at different training batch sizes. The results are in Figure 5. We also measure the training time of GLASS. The results are in Figure 6.

In general, the performance drops as the batch size gets larger. However, the max-zero-one labeling trick is still effective at a large batch size as on all datasets, as the difference of F1 score keeps positive. However, we find that when we set the test and valid batch size to one, the performance of GLASS drops faster, which shows that max-zero-one is not just an approximation of zero-one. With max-zero-one, GLASS is trained to predict the properties of the target subgraph with the existence of other subgraphs.

As the batch size gets larger, the training becomes faster. The time needed for each forward and backward process is nearly irrelevant to batch size, but the gradient is more precise and the model converges faster. For these three datasets, increasing batch size from 1 to 32 reduces training time by about 2/3.

As for other datasets, even plain GNNs can predict component precisely. Furthermore, the targets in real-world datasets are less closely connected to subgraph topologies. Last but not least, real-world

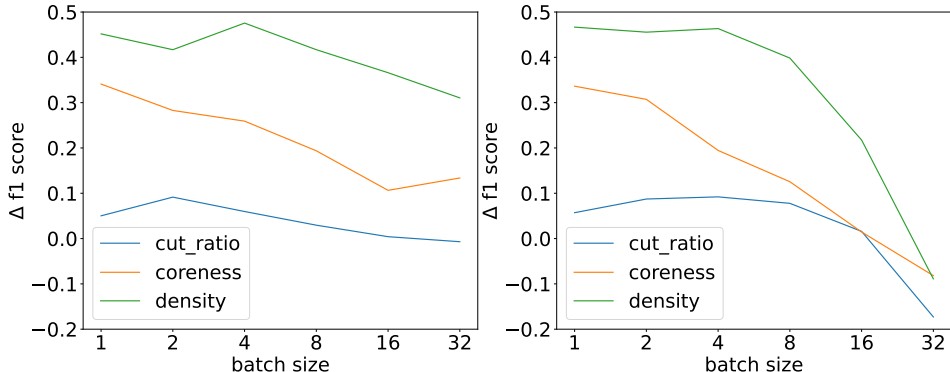

Figure 5: The difference of F1 score between GLASS and GNN-plain. The results are provided with runs on ten random seeds. The left subplot shows the result of testing in batch. The right one shows the performance of testing each sample separately, in other words, setting the valid and test batch size to one.

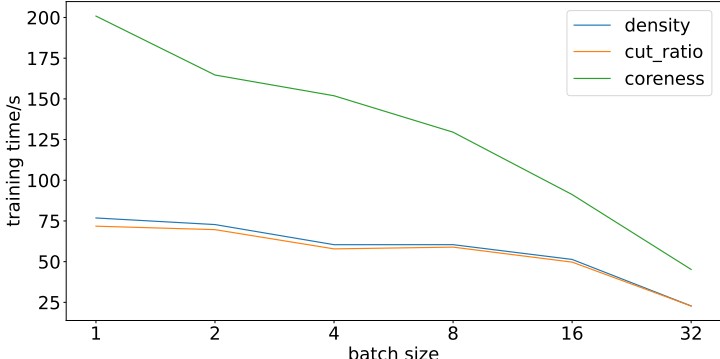

Figure 6: The training time against batch size. The results are provided with runs on ten random seeds.

datasets are bigger, and we also use bigger models for them, leading to large variance in performance and high time complexity.

To trade off between performance and training time, we can tune batch sizes for each dataset. In the worst case when the batch size is so large that target subgraphs in the batch are dense enough to cover the whole graph, GLASS degrades to plain GNN. This process is dataset-related. If target subgraphs are sparsely located in the graph, large batch sizes can be used without hampering the expressive power. Moreover, small batch sizes are not always promising, due to the noise of gradients. We can take a strategy of trying a large batch size first and reducing it until we obtain a model with satisfying expressive power.

