# OpenReview forum: "GLASS: GNN with Labeling Tricks for Subgraph Representation Learning"
_ICLR.cc/2022/Conference — ICLR 2022 Poster_

### Official Review · Reviewer_VqVZ · 2021-10-31

**Correctness:** 3
**Technical Novelty And Significance:** 3
**Empirical Novelty And Significance:** 3
**Recommendation:** 5
**Confidence:** 3

**Main Review:**

The authors study the problem of subgraph classification and propose a simple adaptation of existing GNNs for this task. In particular, their approach augments the initial representation of the node features with label information if a node belongs to a subgraph in a batch.

The paper is clearly written, provides supportive illustrations, and closes theoretical gaps of preceding works.

Authors can improve the paper in the following ways.

1. Provide an ablation study on the self-supervised losses. How much does it affect performance? What if you employ the same losses for other approaches?

2. From the last figure in the appendix it's clear that increasing the batch generally degrades performance, hence the batch size of 1 is the most promising one. What batch size you used in experiments? How much is it a problem from training time perspective? Your approach can't clearly have a very large batch size as it would include all nodes of the graph in the subgraph labels and hence there would be no difference between your approach and plain GNNs. So what's the largest batch size you can afford? What's the trade-off between batch size, time, and performance?

3. Can you please add graph-agnostic MLP and GBDT approaches as baselines for the experiments? They often show superior performance than GNNs (see [here](https://openreview.net/forum?id=ebS5NUfoMKL) for example).

4. One of the biggest concerns, however, is not with the paper, but with a studied problem. While authors give an example where predicting subgraph labels can be useful, I don't believe GNNs should be explored there (it's unlikely that the performance of the department is evaluated based on the graph structure rather on their deliverables). The same is true for the employed datasets, even though they are inherited from the previous work. In particular, ppi-bp, hpo-metab, hpo-neuro ask to predict a label for a subset of proteins, which are obtained from the clinical trials, which are expensive to run and utilize information beyond of what's available in the graphs. Besides the subgraph number of nodes in these datasets is about 10, which are very small graphs to evaluate. em-user is even worse: predicting a gender based on workout history graph is a very contrived dataset and thus it's not very insightful. I understand that it's not the authors who introduced the problem and datasets; however, making it clear why this problem is important and why it cannot be solved by other methods better would significantly strengthen the paper. Maybe you can find some relevant applications in the subgraph regression problem, where your approach could be seamlessly adopted.

I am likely to update my score based on the authors' responses.

**Summary Of The Paper:**

The paper proposes a simple GNN approach to predict labels of subgraphs. In experiments, it works better than an existing approach. Complementary they show their approach can predict characteristics of a graph such as cut ratio. Additionally, they augment the loss with self-supervised losses of predicting node, edge, and subgraph level information.


**Summary Of The Review:**

This is a well-written paper, that could be improved by providing more motivation and exploring the performance in more depth by using other baselines and providing ablation studies.

---

> ### Author Response · Authors · 2021-11-17
> **Author Response 1/2**
>
> We thank the reviewer for their insightful and constructive review.
>
> * Re Q1: "Provide an ablation study on the self-supervised losses. How much does it affect performance? What if you employ the same losses for other approaches?"
>
> We remove our SSLs and the results doesn't vary much. Please see our response to Q2 of reviewer cEit for our new experiment results.
>
> * Re Q2: "From the last figure in the appendix it's clear that increasing the batch generally degrades performance, hence the batch size of 1 is the most promising one. What batch size you used in experiments? How much is it a problem from training time perspective? Your approach can't clearly have a very large batch size as it would include all nodes of the graph in the subgraph labels and hence there would be no difference between your approach and plain GNNs. So what's the largest batch size you can afford? What's the trade-off between batch size, time, and performance?"
>
> Batch sizes used in our datasets are as follows.
>
> | dataset     | density    | cut-ratio     | coreness      | component   |
> | ----------- | ---------- | ------------- | ------------- | ----------- |
> | batch size  | 3          | 3             | 7             | 25          |
> | **dataset** | **ppi-bp** | **hpo-metab** | **hpo-neuro** | **em-user** |
> | batch size  | 80         | 59            | 99            | 6           |
>
> We find that the time needed to train is closely related to the number of forwarding processes per epoch, so we select the batch sizes from $\{\frac{n}{10}, \frac{n}{20}, \frac{n}{40}, \frac{n}{80}\}$, where n is the number of subgraphs in the datasets. This setting can keep the number of forwarding and thus the training time reasonable.
>
> The relation between training time and batch size is shown in Appendix 8, Figure 6.
>
> As the batch size gets larger, the training becomes faster. The time needed for each forward and backward process is nearly irrelevant to batch size, but the gradient is more precise and the model converges faster. For these three datasets, increasing batch size from 1 to 32 reduces training time by about $2/3$.
>
> In the worse case when the batch size is so large that target subgraphs in the batch are dense enough to cover the whole graph, GLASS degrades to plain GNN. This process is dataset-related. If target subgraphs are sparsely located in the graph, large batch sizes can be used without hampering the expressive power. Moreover, small batch sizes are not always promising, due to the noise of gradients. To tradeoff among them, we take a strategy of trying a large batch size first and reducing it until we obtain a model with satisfying expressive power.
>
> Re Q3: "Can you please add graph-agnostic MLP and GBDT approaches as baselines for the experiments? They often show superior performance than GNNs."
>
> We are glad to add such baselines.
>
> Because our datasets provide no node feature, we use pre-trained GNN embeddings for GLASS on real-world datasets and homogeneous input on synthetic datasets. MLP and GBDT on synthetic datasets are meaningless due to homogeneous subgraph embedding. Thus, we only conduct experiments on real-world datasets. We first pool the pre-trained node embeddings to get subgraph embeddings, then use MLP and GBDT (implemented with xgboost) to classify subgraph embeddings.
>
> Results on real-world datasets are as follows. GLASS outperforms MLP and GBDT. Combination of graph topology and node feature is necessary for our datasets.
>
> |           | MLP              | GBDT             | GLASS                   |
> | --------- | ---------------- | ---------------- | ----------------------- |
> | ppi-bp    | $0.445\pm 0.003$ | $0.446\pm 0.000$ | $ \bf{0.619 \pm 0.007}$ |
> | hpo-metab | $0.386\pm 0.011$ | $0.404\pm 0.000$ | $\bf{0.614\pm 0.005}$   |
> | hpo-neuro | $0.404\pm 0.006$ | $0.513\pm 0.000$ | $\bf{0.685\pm 0.005}$   |
> | em-user   | $0.524\pm 0.019$ | $0.694\pm 0.000$ | $\bf{0.888\pm 0.006}$   |

---

> > ### Author Response · Authors · 2021-11-17
> > **Author Response 2/2**
> >
> > 4. Re Q4 : "One of the biggest concerns, however, is not with the paper, but with a studied problem. While authors give an example where predicting subgraph labels can be useful, I don't believe GNNs should be explored there (it's unlikely that the performance of the department is evaluated based on the graph structure rather on their deliverables). The same is true for the employed datasets, even though they are inherited from the previous work. In particular, ppi-bp, hpo-metab, hpo-neuro ask to predict a label for a subset of proteins, which are obtained from the clinical trials, which are expensive to run and utilize information beyond of what's available in the graphs. Besides the subgraph number of nodes in these datasets is about 10, which are very small graphs to evaluate. em-user is even worse: predicting a gender based on workout history graph is a very contrived dataset and thus it's not very insightful. I understand that it's not the authors who introduced the problem and datasets; however, making it clear why this problem is important and why it cannot be solved by other methods better would significantly strengthen the paper. Maybe you can find some relevant applications in the subgraph regression problem, where your approach could be seamlessly adopted."
> >
> > We agree that there are some flaws in the used real-world datasets. However, we would like to respectfully argue that learning representations for substructures bigger than node and edge in the base graph is a general and meaningful problem. In reality, subgraphs can model various phenomenons. Given the structure of a drug molecule, the active spot may contain several atoms and thus form a subgraph. In the anti-money laundering field, recognition of the pattern of complex crime subnetwork is a significant problem. By learning the shareholding structure, we may find some trace of tax evasion. Toxic behavior on social networks also happens among a cluster of people rather than one or two. Subgraph is a far more complex structure than node, edge, and graph, as it has both internal and external topology. To our best knowledge, the existing theory based on graph signal processing or graph isomorphism test can not analyze it directly. Node, edge, and graph are all some specialized cases of subgraphs. The subgraph representation learning problem can give us insight into all these graph representation learning tasks.
> >
> > Though subgraph tasks are very general, subgraph problems are newly raised and existing datasets are mainly designed for node, edge, or graph-level tasks. We will continue finding better datasets to benchmark the subgraph representation learning problem.

---

> > > ### Comment · Reviewer_VqVZ · 2021-11-17
> > > **Re: motivaton**
> > >
> > > I do agree that subgraphs are important structures which are less studied than node-level tasks. However, so far I haven't seen any convincing application/task where existing methods cannot cope with the specified task. For molecules, one needs to conduct lab experiments to determine the properties of the subgraph. For fraud subnetworks there are anomaly detection methods. Can you propose an application case where (1) determining a property of a subgraph is important, (2) GNNs perform well and (3) straightforward/naive solutions are either hard to obtain (e.g. lab experiments) or don't perform well (e.g. by aggregating node-level predictions).

---

> > > > ### Author Response · Authors · 2021-11-17
> > > > **A motivation example in drug discovery**
> > > >
> > > > For molecules, we can indeed use lab experiments to determine subgraph properties. However, lab experiments can be slow and expensive (see [this paper](https://doi.org/10.1093/bioinformatics/btu746) and [this news](https://www.utsouthwestern.edu/newsroom/articles/year-2016/cryo-em-facility.html)), which is why we want to use machine learning to accelerate or replace them. In this context, AlphaFold2 is a great example. As for anomaly detection, many anomaly detection methods exactly utilize GNNs (see [this survey](https://graph-neural-networks.github.io/static/file/chapter26.pdf)). And the previous subgraph representation learning SOTA method SubGNN inspires the design of anomaly detection methods ([this survey](https://arxiv.org/pdf/2106.07178.pdf) gives one example).
> > > >
> > > > In response to the reviewer's asking for a convincing application, we believe our previously mentioned active site prediction task in drug discovery is one. [Active site](https://en.wikipedia.org/wiki/Active_site) is the region of an enzyme where substrate molecules bind and undergo a chemical reaction. Although the active site occupies only ~10–20% of the volume of an enzyme, it is the most important part as it directly catalyzes the chemical reaction. Identification of active sites is crucial in the process of drug discovery, because with identified active site residues we can design drugs which can fit into them. Traditional technologies predict active sites by searching structural similar active sites in the database, which overlooks the fact that active site is bonded to the particular enzyme, and thus should be predicted as a substructure of the enzyme. Modeling it as a subgraph prediction problem is very natural, and a good subgraph representation learning model is expected to reach higher accuracy than traditional matching methods and help the development of new drugs.
> > > >
> > > > Since AI-based drug discovery is a fast-developing new field, we are not sure whether GNN methods are already being studied for active site prediction. But we are aware of a similar work that aims to find a linker molecule (a subgraph) to connect two larger fragments to form a drug [[1]](https://pubs.acs.org/doi/pdf/10.1021/acs.jcim.9b01120). This is a subgraph generation problem conditioned on the given graph, and in principle needs to consider the interaction between subgraph and the rest of the graph. A GNN-based model is developed in [[1]](https://pubs.acs.org/doi/pdf/10.1021/acs.jcim.9b01120) for this task. We are actively generalizing our technique to such subgraph problems in drug discovery.
> > > >
> > > > Furthermore, subgraph representation can also boost other tasks. For example, some works have utilized subgraph representations to help classify graphs (like [this](https://dl.acm.org/doi/10.1145/3442381.3449822)). Motif prediction, a classic graph data mining problem, is also a kind of subgraph classification task. We believe our technique is generally helpful for these tasks.

---

> > > > > ### Comment · Reviewer_VqVZ · 2021-11-18
> > > > > **Re:**
> > > > >
> > > > > The example with active site prediction is not convincing one for the following reasons:
> > > > >
> > > > > (1) There is mostly a single one active site in a molecule. So the training set either contains active site or not, and hence there is no prediction task involved.
> > > > >
> > > > > (2) The subgraphs for active sites are very small, 3-4 nodes. Hardly, GNNs can play any role in this scenario.
> > > > >
> > > > > (3) As subgraphs are very small, it's likely simple methods that aggregate information by nodes work well.
> > > > >
> > > > > (4) Active sites are determined by the 2 molecules in place (enzyme and substrate) and are conducted under specific conditions (e.g. proximity of the molecules, angles, etc.) Topology of the subgraph is not known to play any role for active site prediction.
> > > > >
> > > > > With all that, the datasets you use are different from active site prediction and none of them are convincing to justify the task.
> > > > >
> > > > > While I agree with the authors that AI-based drug discovery is a hot topic right now, I am not yet convinced that subgraph prediction is a valuable task there.

---

> > > > > > ### Author Response · Authors · 2021-11-18
> > > > > > **Active Site Prediction is a Valid Task.**
> > > > > >
> > > > > > The active site prediction task is: "given the structure of an enzyme and another molecule, find out where the binding site is."
> > > > > >
> > > > > > * Re (1): "There is mostly a single one active site in a molecule. So the training set either contains active site or not, and hence there is no prediction task involved."
> > > > > >
> > > > > > We disagree with it.
> > > > > >
> > > > > > Firstly, we do not know why there is no prediction task. Some ML models for this task has already been developed (like  (like DeepSite [1], DoGSiteScorer [2])). Though they do not use GNNs, it illustrates that such task is indeed a meaningful machine learning problem, i.e., the "prediction task" exists.
> > > > > >
> > > > > > Secondly, many enzymes exist with multiple active sites. For example, human fatty acid synthase has six, and cyclosporine synthase has several dozen of active sites. Some enzymes can have several subunits, and each subunit has one active site, like RuBisCO and hetero-oligomer. Moreover, many enzymes have allosteric effects and thus provide different binding sites for control molecules. A common one is ATCase.
> > > > > >
> > > > > > Last but not least, even each enzyme contains only one active set, it does not prevent us from training over multiple enzymes to learn an active site prediction model that can be applied to new enzymes. Note that this does not conflict with our problem definition (small subgraphs locate in a large graph) because the multiple training enzymes can be regarded as components of a large disconnected graph, and the subgraphs to predict each locate in one component. Nevertheless, our point is, generalizing across different graphs (or different parts of the same graph) is a natural ability of GNNs (due to their permutation invariance design and inductive learning ability). So it is not an issue if a graph only contains a single target subgraph given we have multiple graphs to train.
> > > > > >
> > > > > > * Re (2) and (3): "The subgraphs for active sites are very small, 3-4 nodes. Hardly, GNNs can play any role in this scenario." "As subgraphs are very small, it's likely simple methods that aggregate information by nodes work well."
> > > > > >
> > > > > > Firstly, we have clearly described the importance of both internal and external topology for the subgraph representation learning problem. Even if the internal topology is simple, GNNs can still learn from the external topology to help the prediction. Otherwise, GNNs should not work at all for node classification and link prediction tasks (where the internal subgraph contains 1 or 2 nodes). The fact, however, is that GNNs have achieved SoTA performance for both tasks.
> > > > > >
> > > > > > Secondly, enzymes often have complex structure in practice, and we do not agree that active sites are small. Take lactic dehydrogenase as an example, at least amino acid residues Arg$^{171}$, Asp$^{168}$, His$^{195}$, Arg$^{109}$, Gln$^{102}$, Thr$^{246}$, and substrate participate in the reaction. Each one contains at least ten atoms. Modeling these atoms as graph nodes is necessary because otherwise the method may be too coarse. Moreover, the position of these atoms from each other is critical, and the external structure also makes a difference (like steric hindrance). If we simply aggregate node embeddings, we cannot capture the position of atoms relative to other nodes in the active site and the position of the active site relative to other parts of the protein.
> > > > > >
> > > > > > * Re (4): "Active sites are determined by the 2 molecules in place (enzyme and substrate) and are conducted under specific conditions (e.g. proximity of the molecules, angles, etc.) Topology of the subgraph is not known to play any role for active site prediction"
> > > > > >
> > > > > > We agree that active sites have many dependent factors, and the topology of active sites might change case by case. On the one hand, we can put the substrate into the graph. On the other hand, GNNs are known to work well for dynamic graphs. In addition, GNNs can be combined with MD (molecular dynamics) to actively track the change of active site topology. Thus, we do not think this is an issue.
> > > > > >
> > > > > >
> > > > > >
> > > > > > [1] Jiménez, J., Doerr, S., Martínez-Rosell, G., Rose, A. S., & De Fabritiis, G. (2017). DeepSite: protein-binding site predictor using 3D-convolutional neural networks. *Bioinformatics (Oxford, England)*, *33*(19), 3036–3042.
> > > > > >
> > > > > > [2] Volkamer, A., Kuhn, D., Grombacher, T., Rippmann, F., & Rarey, M. (2012). Combining global and local measures for structure-based druggability predictions. *Journal of chemical information and modeling*, *52*(2), 360–372

---

> ### Author Response · Authors · 2021-11-29
> **We are looking forward to your reply**
>
> We thank reviewer VqVZ again for the constructive review and comments to help us improve the paper.
>
> In the initial review, there are four comments. Q1: abalation study on self-supervised learning, Q2: the trade-off between batch size, time and performance, Q3: adding graph-agnostic baselines MLP and GBDT, and Q4: providing more motivation on why subgraph representation learning should be studied and why GNNs are suitable for the task.
>
> We have addressed Q1 in our response to reviewer cEit, addressed Q2 by adding the discussion on batch size (Appendix A.11), and addressed Q3 by adding new experiments using MLP and GBDT (Section 6). Experimental results show that graph-agnostic methods do not perform well on subgraph tasks.
>
> To answer Q4, we tried to list a range of potential subgraph representation learning problems, from active site prediction, network anomaly detection, to motif prediction, molecule linker design, etc. Specifically, we described the active site prediction problem in detail, and answered all the concerns about its validity as a meaningful task and why structure (internal and external) is important for the problem even if subgraph is small.
>
> We would like to sincerely ask whether our responses have addressed reviewer VqVZ's concerns, especially whether they have sufficiently answered Q4. We agree that subgraph representation learning is a new field and there don't exist enough datasets and baselines. However, the subgraph representation learning problem itself is worth studying, not only because many real-world problems can be modeled as subgraphs, but also because subgraph is a general structure to study with much theoretical value (e.g., nodes, links, triplets are all special subgraphs). We believe our work can inspire more research on studying GNN's power for multi-node representation learning, and draw more interest in applying GNNs to real-world subgraph learning problems such as active site prediction and linker design.
>
> We are looking forward to your reply and are more than happy to answer any further questions! Thanks!

---

### Official Review · Reviewer_UU3H · 2021-11-02

**Correctness:** 3
**Technical Novelty And Significance:** 2
**Empirical Novelty And Significance:** 3
**Recommendation:** 6
**Confidence:** 4

**Main Review:**

Strength:

1. The paper addresses the overlooked problem of subgraph-level tasks, as most previous GNN models focus on node or graph-level tasks.

2. The proposed approach is simple and intuitive with little overhead on top of plain GNNs. It can be easily implemented and can be trained efficiently.

3. There is a good balance of theoretical and empirical results.

Weakness:

1. The novelty is somewhat incremental. Node labeling trick has been used on GNNs, although not specifically on the subgraph level. There does not seem to have significant challenges (or rather, it is quite straightforward) in extending plain GNNs to the zero-one / max-zero-one strategies.

2. The main theoretical results only apply to the zero-one strategy. For the max-zero-one strategy, "it can be achieved if the subgraphs are sparsely located in the graph"--- which is dataset dependent. Furthermore, many real-world graphs have a small diameter, meaning that two random subgraph can easily overlapped k-hop neighborhood even when k is a small number.

3. Proposition 2 states that GLASS can cover the six properties of SubGNN. In this case, what is the advantage of GLASS over SubGNN? Any additional property that SubGNN does not have? In the experiments, it is quite clear that GLASS is more efficient, but GLASS also holds a significant accuracy edge. I wonder if this accuracy advantage actually comes from the self-supervised loss, rather than the labeling trick? In particular, in the ablation study, even after removing the labeling trick, the results are still better than SubGNN on the real datasets. In other words, if SubGNN is also trained with the same self-supervised loss, would its performance become much better?

Minor issues:

4. Figure 2 is a bit hard to read, especially for S and S', since the green shapes intersect with many edges. Maybe using two different colors to highlight the nodes/edges, instead of using another shape to enclose it.

5. It is mentioned that to solve label inconsistency in a batch, concatenation cannot be used as it leads to variable lengths of
node labels. I don't quite follow this point. If each batch contain the same number of subgraphs, wouldn't all label vectors have the same length?

--- update after rebuttal ---
Some of the response is valid, especially regarding the self-supervised loss. Hence I'm upgrading my rating.

**Summary Of The Paper:**

In this paper, the authors propose GLASS, a GNN designed for subgraph tasks using labelling tricks. A simple zero-one trick can be used to differentiate nodes in and out of a target subgraphs. However, this would cause efficiency problem as different target subgraphs would require different labels. To fit the classic batch training, a batch of target subgraphs are used to label the nodes together. To resolve conflicts among subgraphs, a zero-max-one trick is proposed. GLASS is easier / intuitive to implement and more scalable. The zero-one trick version is also claimed to have more expressive power than plain GNNs and can capture the six properties in the state-of-the-art method SubGNN. Experiments have been conducted on both real-world and synthetic datasets with generally promising results.

**Summary Of The Review:**

My main reason is incremental novelty (W1), and insufficient support for certain claims (W2/W3).

---

> ### Author Response · Authors · 2021-11-17
> **Author Response 1/2**
>
> We thank the reviewer for their insightful and constructive review.
>
> * Re Q1:  "The novelty is somewhat incremental. Node labeling trick has been used on GNNs, although not specifically on the subgraph level. There does not seem to have significant challenges (or rather, it is quite straightforward) in extending plain GNNs to the zero-one / max-zero-one strategies."
>
> Node labeling tricks have been used in the edge and graph-level tasks primarily in previous works. However, its effectiveness for subgraph-level tasks is verified (both theoretically and empirically) in our paper for the first time. Furthermore, existing labeling tricks have poor scalability due to the relabeling for every target substructure to predict. Our max-zero-one labeling trick provides a new direction of training labeling trick models in batch.
>
> Besides, the existing SOTA method for subgraph prediction (SubGNN) even gives up message passing between nodes and introduces a very complex subgraph-level message passing framework. Our work shows that such a complex subgraph-level message passing is not necessary, and a node-level message passing + labeling trick is enough to outperfrom SubGNN in both accuracy and efficiency. We believe that our work is very timely and important for the subgraph prediction community to focus on GNNs again and to better understand the gap between subgraph prediction and node-level message passing.
>
> * Re Q2: "The main theoretical results only apply to the zero-one strategy. For the max-zero-one strategy, "it can be achieved if the subgraphs are sparsely located in the graph"--- which is dataset dependent. Furthermore, many real-world graphs have a small diameter, meaning that two random subgraph can easily overlapped k-hop neighborhood even when k is a small number."
>
> This concern is also considered during the development of our theoretical analysis. However, as long as we use a batch training strategy, a non-overlapping node labeling trick does not exist. This is proved in Proposition 7. Thus, using a theoretically less powerful labeling trick is a must to balance the expressive power and the computation efficiency. Our solution is the max-zero-one labeling trick which gives label 1 to a node if it appears in at least one subgraph.
>
> Despite being less theoretically powerful, we argue that such a loss of power is acceptable in practice. Firstly, most practical GNNs use a small number of message passing layers, making the overlap much less frequent. For example, we only use 1 or 2 layers in GLASS. Secondly, compared to the average subgraph size, the whole graph size is often much larger (10~100 vs. 20000). Even different subgraphs share some common receptive fields due to a small whole graph diameter, the overlap is not likely to involve a lot of nodes, making most node representations intact. To quantify the overlapping effect, we define *noise ratio* $nr_i$: the ratio of $i$-hop neighbors with inconsistent zero-one and max-zero-one labels.
>
> In our experiment setting, average noise ratios are as follows.
>
> |           | $nr_1$ | $nr_2$ |
> | --------- | ------ | ------ |
> | density   | 0.022  | 0.012  |
> | cut_ratio | 0.018  | 0.012  |
> | coreness  | 0.023  | 0.024  |
> | component | 0.004  | 0.028  |
> | em_user   | 0.018  | 0.011  |
> | ppi_bp    | 0.078  | 0.044  |
> | hpo_metab | 0.063  | 0.040  |
> | hpo_neuro | 0.097  | 0.061  |
>
> As we can see, noisy nodes only takes up a small partion of nodes. Thus, we can reasonably assume that the effect of overlapping node labels is insignificant.

---

> > ### Author Response · Authors · 2021-11-17
> > **Author Response 2/2**
> >
> > * Re Q3(a): "Proposition 2 states that GLASS can cover the six properties of SubGNN. In this case, what is the advantage of GLASS over SubGNN? Any additional property that SubGNN does not have? "
> >
> > Theoretically, GLASS is at least as expressive as SubGNN, and in experiments, GLASS performs better. Moreover, SubGNN manually constructs six channels, but we only need a single message passing channel between nodes. Our framework is more scalable and straightforward. Last but not least, the units of SubGNN utilize subgraph patches sampled from the whole graph randomly, leading to high variance in performance and dubious robustness.
> >
> > * Re Q3(b): "In the experiments, it is quite clear that GLASS is more efficient, but GLASS holds a significant accuracy edge. I wonder if this accuracy advantage actually comes from the self-supervised loss, rather than the labeling trick? In particular, in the ablation study, even after removing the labeling trick, the results are still better than SubGNN on the real datasets. In other words, if SubGNN is also trained with the same self-supervised loss, would its performance become much better?"
> >
> > We decide to remove the SSLs. Please see our response to Q2 of reviewer cEit for our new experiment results. GLASS still outperforms all of our baselines.
> >
> > For completeness, we also conduct experiments by adding our SSL losses to SubGNN. SSLs seem to degrade the performance of SubGNN. Experiment results are as follows. They indicate that our SSL tasks are not compatible with SubGNN, and the main advantage of GLASS comes from the labeling trick.
> >
> > |            | hpo_neuro             | em_user               | ppi_bp          | hpo_metab        |
> > | ---------- | --------------------- | --------------------- | --------------- | ---------------- |
> > | SubGNN+SSL | $0.606\pm 0.004$      | $0.806\pm 0.013$      | *               | *                |
> > | SubGNN     | $\bf{0.644\pm 0.006}$ | $\bf{0.816\pm 0.013}$ | $0.599\pm 0.08$ | $0.537\pm 0.006$ |
> >
> > *: get stuck on long precomputation (more than 24 h).
> >
> > * Re Q4: "Figure 2 is a bit hard to read, especially for S and S', since the green shapes intersect with many edges. Maybe using two different colors to highlight the nodes/edges, instead of using another shape to enclose it."
> >
> > We are glad to fix it.
> >
> > * Re Q5: "It is mentioned that to solve label inconsistency in a batch, concatenation cannot be used as it leads to variable lengths of node labels. I don't quite follow this point. If each batch contain the same number of subgraphs, wouldn't all label vectors have the same length?"
> >
> > Yes, fixed batch sizes lead to fixed-length label vectors. However, inconsistency means different labels of nodes in target subgraphs in other batches(even the same batch in another order). For example, assuming batch size = 2, the label of one node can be $[0, 1]$ or $[1,0]$ according to the order of subgraphs within the batch. Such inconsistency makes the model no longer permutation invariant, which violates the fundamental design principle of GNNs and will significantly worsen the sample complexity.

---

> ### Author Response · Authors · 2021-11-29
> **We are looking forward to your reply**
>
> We thank reviewer UU3H again for the constructive review and comments to help us improve the paper.
>
> There are five concerns in the review. Q1: the novelty of our method, Q2: whether our approximation (the max-zero-one labeling trick) is valid for our datasets, Q3: how much SSL contributes to the performance of GLASS, Q4: format of Figure 2, and Q5: the meaning of label inconsistency.
>
> We have addressed Q1 by summarizing our contribution in our reply, addressed Q2 by quantifying the overlapping effect in Appendix A.9, and addressed Q3 in our response to reviewer cEit. Experimental results show that SSL is not a major reason for the success of our model and the overlapping effect is insignificant. In response to Q4 and Q5, we fixed Figure 2 in our paper and clarified the meaning of label inconsistency in the reply.
>
> We were wondering whether our response and revision have sufficiently addressed your concerns, and whether you are satisfied with our added experiments. We are looking forward to your reply!

---

### Official Review · Reviewer_8Ehn · 2021-11-02

**Correctness:** 3
**Technical Novelty And Significance:** 2
**Empirical Novelty And Significance:** 2
**Recommendation:** 6
**Confidence:** 4

**Main Review:**

Strengths:
1) This work is well-motivated. The authors claim that distinguishing inside and outside nodes of subgraphs is crucial for subgraph tasks. Based on the insight, the authors propose a novel method for node-level message passing GNNs to perform subgraph representation learning effectively and efficiently.
2) The authors theoretically analyze the effectiveness of the max-zero-one labeling trick and the expressiveness power of the proposed method.
3) Experiments demonstrate that the proposed model with labeling tricks outperforms state-of-the-art subgraph representation learning methods on several synthetic and real-world datasets. Experiments on running time also show that the proposed model is more computationally efficient than the subgraph-level learning method SubGNN.

Weakness:
The authors may want to improve the presentation of this paper. For example, the introduction and analysis of SubGNN take up too much space that the illustration of the proposed method and experimental settings are not sufficiently detailed in the main text.


**Summary Of The Paper:**

This paper focuses on predicting the properties of subgraphs in the whole graph. The authors propose a labeling trick to help GNN distinguish nodes inside and outside the subgraph. The authors rigorously analyze the effectiveness of the proposed method. Experiments demonstrate that the proposed method achieves state-of-the-art performance on several benchmarks.

**Summary Of The Review:**

The authors propose a labeling trick to enhance plain GNNs for subgraph tasks, and the theoretical and empirical results demonstrate its effectiveness and efficiency. However, the authors may want to detail the model structure and experimental settings in the main text.

---

> ### Author Response · Authors · 2021-11-17
> **Author Response**
>
> We thank the reviewer for constructive review. We will add more details on the model architecture and experiments on it, and further improve the presentation in the revised paper.

---

### Official Review · Reviewer_cEit · 2021-11-04

**Correctness:** 3
**Technical Novelty And Significance:** 2
**Empirical Novelty And Significance:** 2
**Recommendation:** 6
**Confidence:** 4

**Main Review:**

This paper argues that a subgraph-level positional encoding is enough for subgraph representation learning. It gives some theoretical analysis that such trick could fulfill the six properties that subGNN proposed. It also introduces three self-supervised learning task to guide GNN learning.

I think the approach is reasonable and the analysis is sound and convincing. The main concern is:

1) This paper adds the self-supervised learning into the framework, and compare with subGNN without such SSL learning. I think this comparison is not fair. On the one hand, I think SSL learning should be agnostic to GNN model. I think you should compare your model with other GNN model in the same setting, i.e., with SSL training or without.
2) The motivation and insights of the SSL learning task is less than the subgraph encoding. I think the authors could consider either remove this part, or at least compare with other SSL learning methods, such as graph contrastive learning or generative learning.
3) The idea of adding structural encoding is also explored in graph transformer (Do Transformers Really Perform Bad for Graph Representation?)
I recommend the authors also compare this method into subgraph-level modelling.



**Summary Of The Paper:**

This paper proposes a simple subgraph modelling framework by adding a subgraph-level position encoding to node (whether a node belongs to subgraph) to node feature. In this way, the GNN is able to differentiate nodes from different subgraphs, and outperform previous model SubGNN.

**Summary Of The Review:**

The method is simple and effective, but I doubt the current evaluation setup as it uses additional self-supervision to train the model against baselines.

---

> ### Author Response · Authors · 2021-11-17
> **Author Response 1/2**
>
> We thank the reviewer for their comprehensive and constructive review.
>
> * Re Q2: "The motivation and insights of the SSL learning task is less than the subgraph encoding. I think the authors could consider either remove this part, or at least compare with other SSL learning methods, such as graph contrastive learning or generative learning."
>
> What inspires us to use SSL tasks is that our datasets provide no node feature. A straightforward way to produce input for GNN is to use node degree. However, a better method (also utilized by SubGNN) is to use the embeddings produced by GNN pre-trained on link prediction (edge-level SSL) tasks as the input node features. To generate node embeddings capturing node and subgraph-level task information, we introduce node and subgraph SSL tasks.  Moreover, pre-trained GNN node embeddings (pre-trained on the edge and node-level SSL tasks) also help GLASS capture distant neighbors. If the pre-trained GNN has $l$ layers and the GNN used in GLASS has $k$ layers, it can capture $(l + k)$-hop neighbors, while a $k$-layer MPNN can only capture neighbors within $k$ hops.
>
> We agree that the SSL part is a minor contribution, which is orthogonal to our main technique, including it might obscure the core idea of our paper. Therefore, we decide to remove it from our paper and use the same method as in SubGNN to produce node features.
>
> * Re Q1: "This paper adds the self-supervised learning into the framework, and compare with subGNN without such SSL learning. I think this comparison is not fair. On the one hand, I think SSL learning should be agnostic to GNN model. I think you should compare your model with other GNN model in the same setting, i.e., with SSL training or without."
>
> We present the new experimental results on real-world datasets as follows.
>
> |           | SubGNN           | GLASS                     | GNN-plain         | GLASS+SSL             |
> | --------- | ---------------- | ------------------------- | ----------------- | --------------------- |
> | ppi-bp    | $0.599\pm 0.008$ | $0.619\pm 0.007$          | $0.613\pm  0.006$ | $\bf{0.621\pm 0.008}$ |
> | hpo-metab | $0.537\pm 0.006$ | $\mathbf{0.614\pm 0.005}$ | $0.597\pm 0.012$  | $0.565\pm 0.006$      |
> | hpo-neuro | $0.644\pm 0.006$ | $\mathbf{0.685\pm 0.005}$ | $0.668\pm 0.007$  | $0.674\pm 0.002$      |
> | em-user   | $0.816\pm 0.013$ | $0.888\pm 0.006$          | $0.847 \pm 0.021$ | $\bf{0.902\pm 0.006}$ |
>
> As we can see, with the same node feature generation method, GLASS still outperforms SubGNN and GNN-plain. To our surprise, we observe that although GLASS's performance drops a bit on ppi-bp and em-user, its performance increases on hpo-metab and hpo-neuro. This again verifies that the SSL part is not essential in GLASS---the main advantage lies in the zero-one labeling trick + GNN.
>
> Note that the experiments on synthetic datasets did not use SSL originally, thus do not need to be rerun. In addition, the baseline GNN-seg learns subgraph embedding without information outside the subgraph. Using SSL may leak the outside topology to it. The other baseline Sub2Vec is already an SSL method (like node2vec) to generate subgraph embeddings without taking node embeddings as input. So we do not modify these two baselines.
>
> This shows that the usefulness of SSL is actually dependent on datasets. Since we decide to remove SSL from the current paper, we leave the exploration of different SSL techniques for different datasets to future work.

---

> > ### Author Response · Authors · 2021-11-17
> > **Author Response 2/2**
> >
> > * Re Q3: "The idea of adding structural encoding is also explored in graph transformer (Do Transformers Really Perform Bad for Graph Representation?) I recommend the authors also compare this method into subgraph-level modelling."
> >
> > We agree that Graphormer is excellent work. However, The problem of applying Graphormer directly to the whole graph is that Graphormer uses several tensors with shape $n_{\text{head}}\times|V|\times |V|$, where $V$ is the node set and $n_{\text{head}}$ is the number of attention heads. If we use Graphormer ($n_{\text{head}}=16$) on the whole graph(~20000 nodes), such one tensor takes 12 GB GPU memory. And applying Graphormer on subgraph tasks is not very straightforward, as it utilizes virtual nodes rather than pooling node embeddings to produce high-order structure embedding. Therefore, we treat it as a graph-level model by applying it to subgraphs segregated from the whole graph and compare it with GNN-seg.
> >
> > We use 2 layers and 16 heads. Hidden dimension is 16 for synthetic datasets and 64 for real-world datasets. We also reduce the tot_updates to 3000 and warmup_updates to 120.  All other hyperparameters are the same as in the default setting.
> >
> > |           | GNN-seg           | Graphormer            | GLASS                     |
> > | --------- | ----------------- | --------------------- | ------------------------- |
> > | density   | $0.952 \pm 0.006$ | $0.932\pm 0.009$      | $\bf{0.956\pm 0.006}$     |
> > | cut ratio | $0.346 \pm 0.011$ | $0.362\pm 0.021$      | $\bf{0.949\pm 0.009}$     |
> > | coreness  | $0.593 \pm 0.012$ | $\bf{0.851\pm 0.017}$ | $0.840\pm 0.009$          |
> > | component | $1.000\pm 0.000$  | $1.000\pm 0.000$      | $1.000\pm 0.000$          |
> > | ppi_bp    | $0.361\pm 0.008$  | $0.372\pm 0.005$      | $\bf{0.619\pm 0.007}$     |
> > | hpo_metab | $0.542\pm 0.009$  | $0.521\pm 0.004$      | $\mathbf{0.614\pm 0.005}$ |
> > | hpo_neuro | $0.647\pm 0.001$  | $0.491\pm 0.004$      | $\mathbf{0.685\pm 0.005}$ |
> > | em_user   | $0.725\pm 0.003$  | Out of Memory         | $\bf{0.888\pm 0.006}$     |
> >
> > As we can see, the performance of Graphormer is similar to GNN-seg, and in general worse than GLASS except for the systhetic dataset coreness. We suspect that the capacity of Graphormer is so large that it can capture some statistical relation between inner and outside topology by remembering the data generation pattern. We will add the Graphormer experiments in the revised paper.

---

> > > ### Comment · Reviewer_cEit · 2021-11-17
> > > **Clarification about my concern about structural encoding**
> > >
> > > Thanks the authors for the comprehensive reply.
> > >
> > > 1) I'm glad to see that even without SSL component, the structural encoding for subgraph still gives promising results, as it's simple and clean.
> > >
> > > 2) Regarding to the comparison with Graphformer, my point is to compare "just" with the structural encoding method they utilize, i.e., centrality and spatial encoding. As if your main contribution is the structural encoding for subgraph, it would be great to summarize all the existing works about structural encoding, and then discuss and prove their limitation when applied to subgraph.
> > >
> > > I think you could refer to section 5.2 in Graphformer paper to see more works about structure encoding on graph, and probably discuss your novelty against them.

---

> > > > ### Author Response · Authors · 2021-11-18
> > > > **Response 1/2**
> > > >
> > > > Thank you for this suggestion. We will add this discussion to the revised paper. Some preliminary discussion is as follows.
> > > >
> > > > In Graphormer, the centrality encoding is a function of indegree and outdegree. Though it is powerful, such encoding is common for plain GNNs: they usually use one-hot degree as input node features and the first linear layer can be considered as embeddings for the degree. We also use it for pre-trained GNN to produce node embeddings. Moreover, even with homogenuous input without any information, GIN can easily learn the node degree with one layer. We believe centrality encoding is specifically useful for graph transformers, as these models treat any graph structure as a complete graph which loses the node degree information.
> > > >
> > > > As for the spatial encoding of Graphormer, it is orthogonal to our zero-one node labeling trick as the spatial encoding cannot differentiate nodes inside and outside the target subgraph, which is proved to be very important for subgraph representation learning. So is the edge encoding method.
> > > >
> > > > We discuss other existing structural encoding methods as follows.
> > > >
> > > > **Path and Distance in GNNs**. [Spagan](https://arxiv.org/pdf/2101.03464.pdf) [1] and [PAGTN](https://arxiv.org/pdf/1905.12712.pdf) [2] both utilize shortest path encodings to reconstruct the adjacency matrix. They can capture long-range node interaction, which is important for subgraph tasks, but orthogonal to our zero-one node labeling tricks to distinguish nodes inside the target subgraph. [P-GNN](http://proceedings.mlr.press/v97/you19b/you19b.pdf) [3] can capture the position of nodes relative to the graph but not to the target subgraph, and is not inductive due to the positional node feature. [Distance Encoding](https://arxiv.org/pdf/2009.00142.pdf) [4] can capture the position of nodes relative to a target node set. But as these two methods compute the distance from all nodes to target nodes, they are not scalable for our datasets. Moreover, the latter disables batch training.
> > > >
> > > > **Positional Encoding in Graph Transformers**. [WL-PE](https://arxiv.org/pdf/2001.05140.pdf) [5] utilizes the Weisfeiler-Lehman algorithm to produce node labels. As GNN can be as expressive as the WL test, we do not think GNN with this technique can be made more expressive. From the design of node labeling tricks, [Laplacian-PE](https://arxiv.org/pdf/2012.09699.pdf) [6] is neither invariant nor expressive, as isomorphic nodes can have different labels, while non-isomorphic nodes can have the same label (for example the eigenvector corresponding to eigenvalue 0).
> > > >
> > > > **Edge Feature**. They are not designed for subgraph tasks and can not represent node position relative to the target subgraph. And we can easily add our node labeling tricks to [EGNN](https://openaccess.thecvf.com/content_CVPR_2019/papers/Gong_Exploiting_Edge_Features_for_Graph_Neural_Networks_CVPR_2019_paper.pdf) [7], [GINE+](https://arxiv.org/pdf/2011.15069.pdf) [8].
> > > >
> > > > **Other Node Labeling Tricks**: Please refer to the Related Work section in our paper for these works. Our paper verifies the effectiveness of node labeling tricks for subgraph-level tasks for the first time. Furthermore, most existing labeling tricks have poor scalability while our max-zero-one node labeling trick enables batch training.

---

> > > > > ### Author Response · Authors · 2021-11-18
> > > > > **Response 2/2**
> > > > >
> > > > > We also conduct an experiment on replacing max-zero-one label with the centrality encoding method of Graphormer. Results shows that centrality encoding is not effective for subgraph tasks.
> > > > >
> > > > > |           | GNN+Centrality Encoding | GNN-plain              | GLASS                     |
> > > > > | --------- | ----------------------- | ---------------------- | ------------------------- |
> > > > > | density   | $\it{0.662\pm 0.010}$   | $0.470\pm 0.006$       | $\bf{0.956\pm 0.006}$     |
> > > > > | cut ratio | $0.616\pm 0.008$        | $\it{0.887\pm 0.009}$  | $\bf{0.949\pm 0.009}$     |
> > > > > | coreness  | $0.460\pm  0.016$       | $\it{0.689\pm 0.012}$  | $\bf{0.840\pm 0.009}$     |
> > > > > | component | $0.644\pm 0.036$        | $\it{0.998\pm 0.002}$  | $\bf{1.000\pm 0.000}$     |
> > > > > | ppi_bp    | $0.580\pm 0.005$        | $\bf{0.613\pm  0.006}$ | $\bf{0.619\pm 0.007}$     |
> > > > > | hpo_metab | $\it{0.603\pm 0.010}$   | $0.597\pm 0.012$       | $\mathbf{0.614\pm 0.005}$ |
> > > > > | hpo_neuro | $\it{0.671\pm 0.008}$   | $0.668\pm 0.007$       | $\mathbf{0.685\pm 0.005}$ |
> > > > > | em_user   | $\it{0.861\pm 0.012}$   | $0.847 \pm 0.021$      | $\bf{0.888\pm 0.006}$     |
> > > > >
> > > > > We can see that centrality encoding can boost plain GNNs on some datasets, but it is can not differentiate nodes inside and outside, leading to low score on density and cut ratio dataset.
> > > > >
> > > > > As for the path and edge encoding, they both takes $O(|V|^2)$ space and at least $O(|V|^2)$ time complexity per forward process, we can not afford them.
> > > > >
> > > > >
> > > > >
> > > > > [1] Yiding Yang, Xinchao Wang, Mingli Song, Junsong Yuan, and Dacheng Tao. Spagan: Shortest path graph attention network. Advances in IJCAI, 2019.
> > > > >
> > > > > [2] Benson Chen, Regina Barzilay, and Tommi Jaakkola. Path-augmented graph transformer network. arXiv preprint arXiv:1905.12712, 2019.
> > > > >
> > > > > [3] Jiaxuan You, Rex Ying, and Jure Leskovec. Position-aware graph neural networks. In International Conference on Machine Learning, pages 7134–7143. PMLR, 2019.
> > > > >
> > > > > [4] Pan Li, Yanbang Wang, Hongwei Wang, and Jure Leskovec. Distance encoding: Design provably more powerful neural networks for graph representation learning. Advances in Neural Information Processing Systems, 33, 2020.
> > > > >
> > > > > [5] Jiawei Zhang, Haopeng Zhang, Congying Xia, and Li Sun. Graph-bert: Only attention is needed for learning graph representations. arXiv preprint arXiv:2001.05140, 2020.
> > > > >
> > > > > [6] Vijay Prakash Dwivedi and Xavier Bresson. A generalization of transformer networks to graphs. AAAI Workshop on Deep Learning on Graphs: Methods and Applications, 2021.
> > > > >
> > > > > [7] Liyu Gong and Qiang Cheng. Exploiting edge features for graph neural networks. In Proceedings of the IEEE/CVF Conference on Computer Vision and Pattern Recognition, pages 9211–9219, 2019.
> > > > >
> > > > > [8] Rémy Brossard, Oriel Frigo, and David Dehaene. Graph convolutions that can finally model local structure. arXiv preprint arXiv:2011.15069, 2020.

---

> > > > > > ### Comment · Reviewer_cEit · 2021-11-21
> > > > > > **Convinced about the response**
> > > > > >
> > > > > > I'm convinced about the authors' response, and decided to raise the score to 6.
> > > > > >
> > > > > > If given more time, it would be better that the authors could briefly discuss what could be other potential approaches that encode subgraph information.

---

> > > > > > > ### Author Response · Authors · 2021-11-22
> > > > > > > **Thank You**
> > > > > > >
> > > > > > > Thank you for your suggestions for this paper. We will continue working on the subgraph representation learning problem.

---

### Author Response · Authors · 2021-11-17
**Summary of Additional Experiments**

Dear reviewers,

We greatly appreciate your effort in reviewing our paper. Below we make a brief summary of the additional experiments we added during the author response period.

In response to the common concern about the unfair advantage Self-Supervised-Learning (SSL) brings to our model, we add an experiment by removing the SSL from our model. Please see our response to Q2 of reviewer cEit for the new experimental results. The new results show that without SSL, our GLASS model still significantly outperforms the baselines, and the main advantage of our model lies in the zero-one labeling trick rather than the SSL. We also try to equip the baselines with our SSL technique. The experimental results are included in our response to Q3(b) of reviewer UU3H.

Moreover, reviewers are concerned about the batch sizes used in our experiments. We list the batch sizes in our response to Q2 of reviewer VqVz. We also illustrate the relation between training time and batch size there. To ease the concern of reviewer UU3H that subgraphs can easily overlap, we introduce a score to measure how subgraphs and their neighbors overlap. Please see our response to Q2 of reviewer UU3H for these scores on our datasets. Under our experimental settings, we can reasonably assume that the effect of overlapping node labels is insignificant.

Last but not least, reviewer cEit recommends a new baseline Graphormer. We also add non-graph baselines MLP and GBDT based on the suggestion from reviewer VqVz. Please see our response to Q3 of reviewer cEit and Q3 of reviewer VqVz for these experimental results. Our model outperforms all these baselines.

We sincerely hope the added results can alleviate your concerns about our paper, and we are more than happy to answer any further questions raised during the author response. Please see our detailed response to each reviewer in their respective section.

Thanks,

Authors

---

### Author Response · Authors · 2021-11-21
**Summary of Modifications to the Paper.**

Dear reviewers,

Below we make a brief summary of our modifications to the paper.

In response to the common concern about the unfair advantage Self-Supervised-Learning (SSL) brings to our model, we decide to remove SSL from our main paper and move its results and discussion to Appendix A.8. We also update the experimental results of GLASS and GNN-plain on real-world datasets after removing the SSL.

In response to reviewer cEit, we add a comparison between our max-zero-one node labeling trick with existing structural encoding methods in Appendix A.1.

In response to reviewer 8Ehn, we introduce the architecture and experimental settings of GLASS in Section 6.

In response to reviewer UU3H, we quantify how much targets subgraphs overlap with each other in Appendix A.9. We also draw a new Figure 2.

In response to reviewer VqVz, we add GBDT and MLP as new baselines, whose results can be found in Section 6. We also discuss the relationship between batch size and training time and how to trade off between training time and performance in Appendix A.11.

Thanks,

Authors

---

### Decision · Program_Chairs · 2022-01-20

**Decision:**

Accept (Poster)

**Comment:**

This paper proposes a labeling trick for subgraph representation learning with GNNs. The proposed method, GLASS, improves on subgraph-level tasks. The topic of subgraph representation learning is relatively new, and this paper makes progress in that community which would be appreciated by other researchers interested in the same problem.

The paper in the original submission state raised some concerns from the reviewers about unclear writing of the motivation and potential applications, technical novelty, and comparisons with existing approaches (even one that are not specifically designed for subgraph representation learning). It is good that the authors conducted additional experiments to show the effect of SSL (that the approach makes improvements without SSL). This and other clarifications from the authors convinced the reviewers to recommend acceptance.